# ProtComposer: Compositional Protein Structure Generation with 3D Ellipsoids

**Hannes Stark**[*,1,2] , **Bowen Jing**[*,1,2], **Tomas Geffner**[1], **Jason Yim**[2], **Tommi Jaakkola**[2], **Arash Vahdat**[1] **& Karsten Kreis**[1]

[1]NVIDIA, [2]CSAIL, Massachusetts Institute of Technology

## Abstract

We develop ProtComposer to generate protein structures conditioned on spatial protein layouts that are specified via a set of 3D ellipsoids capturing substructure shapes and semantics. At inference time, we condition on ellipsoids that are hand-constructed, extracted from existing proteins, or from a statistical model, with each option unlocking new capabilities. Hand-specifying ellipsoids enables users to *control* the location, size, orientation, secondary structure, and approximate shape of protein substructures. Conditioning on ellipsoids of existing proteins enables redesigning their substructure's connectivity or *editing* substructure properties. By conditioning on novel and diverse ellipsoid layouts from a simple statistical model, we improve protein generation with expanded Pareto frontiers between designability, novelty, and diversity. Further, this enables sampling designable proteins with a helix-fraction that matches PDB proteins, unlike existing generative models that commonly oversample conceptually simple helix bundles. Code is available at https://github.com/NVlabs/protcomposer.

## 1 Introduction

Proteins are intricate macromolecular machines that carry out a wide variety of biological and chemical processes. A grand vision of rational protein design is to be able to design complex and modular functions akin to those found in nature, where different spatial parts of the protein possess different properties that act in a coordinated fashion (Chu et al., 2024; Kortemme, 2024). However, current paradigms of ML-based protein structure generation are largely limited to unconditional generation (Watson et al., 2023; Yim et al., 2023b), or to inpainting of scaffolds and binders conditioned on known parts of the structure (Watson et al., 2023; Trippe et al., 2022), with no ability to control the higher-level spatial placement or layout of the generated protein. This leads to limited diversity and control of the generated samples and distinguishes protein generation from image generation, where such levels of control are commonplace and lead to new capabilities in the hands of human users (Li et al., 2023a; Rombach et al., 2022; Nie et al., 2024; Zheng et al., 2023a; Zhang et al., 2023a).

To bridge this gap in the protein design toolbox, we develop ProtComposer as a means of controlling protein structure generative models with the protein's layout in 3D space. Specifically, we describe modular protein layouts via *3D ellipsoids* augmented with annotations to provide a rough "sketch" of the protein (Figure 1). Similar to blob or bounding box representations in image generation (Nie et al., 2024; Li et al., 2023a), these ellipsoids provide a level of abstraction intermediate between data-level (i.e., voxels) constraints and global conditioning. They are informative enough to control the generation of diverse proteins, but are human-interpretable, easy-to-construct, and do not constrain the low-level details of protein structures. Hence, 3D ellipsoids facilitate a two-stage paradigm in which complex protein designs are expressed as spatial sketches by hand or via heuristic algorithms, and deep learning models "fill in" these sketches with high-quality, designable backbones.

In this work, we apply our philosophy to controlling Multiflow (Campbell et al., 2024), a joint sequence-structure flow-matching model with state-of-the-art designability, with 3D ellipsoid layouts annotated with **secondary structure**. Multiflow represents protein structures as a cloud of residue frames in $SE(3)$ and parameterizes the flow network with Invariant Point Attention (Jumper

---

[*]Equal contribution. Work done during internships at NVIDIA. {hstark, bjing}@mit.edu.

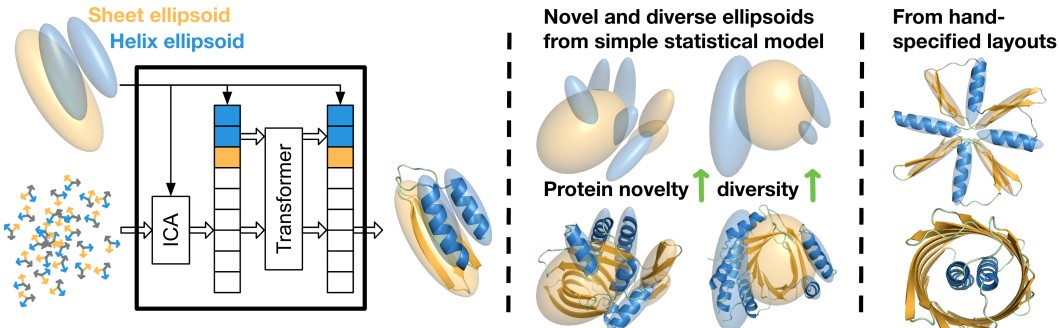

Figure 1: _Left:_ ProtComposer's flow model denoises a noisy protein structure conditioned on a spatial protein layout of annotated ellipsoids that we inject using our Invariant-Cross Attention (ICA) and ellipsoid tokens. _Middle:_ At inference time, we condition our model on ellipsoids from a simple statistical model (Sec. 3.4) that guarantees novel and diverse layouts. The generated proteins exhibit high novelty and diversity while maintaining designability. _Right:_ Conditioning on hand-specified ellipsoids allows for controllable generation.

et al., 2021; Yim et al., 2023b). To inject ellipsoid conditioning into this network, we develop an equivariant mechanism for message passing between ellipsoids in 3D space and residue frames, which we call _Invariant Cross Attention_. We then fine-tune Multiflow with this cross attention into a conditional model of protein structure and sequence, conditioned on ellipsoid layouts. We develop a classifier-free guidance mechanism with Multiflow, enabling interpolation between ellipsoid-conditioned and unconditional generation. Empirically, we find this family of conditional generative distributions advances the state of the art in protein generation along three axes:

- **Control**—unlike existing models, we can prompt our method on (existing or novel) arrangements of secondary structure ellipsoids. We develop a family of metrics to measure adherence to ellipsoid conditioning and find strong consistency between conditioning ellipsoid layouts and generated backbones. This consistency persists well beyond the training distribution, with some particularly impressive generations shown in Figure 7.

- **Diversity and Novelty**—by conditioning on synthetic ellipsoid layouts drawn from a family of simple statistical models (Section 3.4), we significantly increase the diversity and novelty of generations. Although there is a cost to the designability of the generated proteins, our Pareto frontier along this tradeoff far surpasses that of adjusting inference-time parameters, the only existing option for controlling the diversity of Multiflow generations.

- **Compositionality**—although highly designable, protein generations often exhibit a low degree of architectural complexity (e.g., only composed of a single alpha-helix bundle). We argue that such proteins are analogous to pathological language model outputs with high model likelihood but low information content (i.e., the sentence "`and and and ...`" has high likelihood under many large language models.). We introduce a _compositionality_ metric to quantify this phenomenon and show that ellipsoid conditioning can improve the complexity and compositionality of generated structures.

## 2 BACKGROUND AND RELATED WORK

**Protein Structure Generation.** The primary aim of protein structure generative models (Yim et al., 2023b; Ingraham et al., 2023; Bose et al., 2024; Campbell et al., 2024) is aiding computational design of _novel_ proteins (Watson et al., 2023; Lauko et al., 2024). Thus, we often desire to generate _beyond already existing_ folds or secondary structure compositions, and to _control_ generations to satisfy design specifications. To address controllability, there are two main explored avenues beyond scaffolding existing structural motifs (Yim et al., 2024a). First, conditioning on block contact maps and sequential secondary structure specifications (Anand & Achim, 2022; Watson et al., 2023). Second, _inference time_ conditioning, as in Chroma (Ingraham et al., 2023), via projections onto a manifold or via forces from an arbitrary differentiable energy function. Such inference time control enjoys high generality while ProtComposer is trained for a single type of shape and semantic conditioning, but, therefore, enjoys improved adherence to the control (Table 1).

**Spatial Conditioning.** For image generative models, controllable generation has enabled new applications that make up a significant portion of their utility beyond generating impressive images. To unlock similar new capabilities for protein generation, we follow the transferrable concepts that have crystallized out as crucial. On a technical and architectural level, this includes fine-tuning strong existing generative models with the principle of minimally perturbing the original model's output in the initialized conditional model (Li et al., 2023a; Zhang et al., 2023b). On a conceptual level, this implies finding the right input specification: for different tasks, different levels of granularity are appropriate. In the image domain, this ranges from pixel-level specifications such as semantic segmentation maps or sketches to more coarse-grained specifications such as bounding-boxes (Li et al., 2023b; Zhang et al., 2023b), or "blobs" (Nie et al., 2024), akin to our aims for proteins.

**Flow models.** Flow matching (Liu et al., 2022; Lipman et al., 2022; Albergo & Vanden-Eijnden, 2022; Albergo et al., 2023) aims to learn a time-dependent vector field $v_{\theta,t}$ that, when integrated from a start time $t = 0$ to $t = 1$, transports samples from a noise distribution $\mathbf{x}_0 \sim p_0$ to a data distribution $\mathbf{x}_1 \sim p_1$. To train $v_{\theta,t}$, we sample partially noised data from a *conditional probability path* $p_t(\mathbf{x} \mid \mathbf{x}_0, \mathbf{x}_1)$ satisfying $p_0(\mathbf{x} \mid \mathbf{x}_0, \mathbf{x}_1) \approx \delta(\mathbf{x} - \mathbf{x}_0)$ and $p_1(\mathbf{x} \mid \mathbf{x}_0, \mathbf{x}_1) \approx \delta(\mathbf{x} - \mathbf{x}_1)$. A common choice is a Dirac that traces out a straight line between $\mathbf{x}_0$ and $\mathbf{x}_1$ or a geodesic for flow matching on manifolds (Chen & Lipman, 2024). At the sampled noisy datapoints $\mathbf{x}_t$, we evaluate the vector field $v_{\theta,t}(\mathbf{x}_t)$ and regress it against the *conditional vector field* $u_t(\mathbf{x}_t \mid \mathbf{x}_0, \mathbf{x}_1)$ that corresponds to the conditional probability path through the continuity equation $\frac{\partial}{\partial t} p_t = -\nabla \cdot (p_t u_t)$. At convergence, $v_{\theta,t}$ approximates the *marginal vector field* $u_t(\mathbf{x})$ (since the gradients are equivalent to regressing against $u_t(\mathbf{x})$) that evolves the prior $p_0$ to the data distribution $p_1$ through the *marginal probability path* $p_t(\mathbf{x}) = \int p_t(\mathbf{x} \mid \mathbf{x}_0, \mathbf{x}_1) p_0(\mathbf{x}_0) p_1(\mathbf{x}_1) d\mathbf{x}_0 d\mathbf{x}_1$.

## 3 METHOD

### 3.1 ELLIPSOID REPRESENTATION OF PROTEINS

Proteins are compositional objects—different regions have different properties, and we seek a language in which to succinctly describe this information to control the sampling of a generative model. To do so, we propose to represent a protein's spatial layout using a set of $K$ ellipsoids, each corresponding to a semantically coherent region of the protein. Each ellipsoid records the *number* of residues in the associated region, a categorical *semantic feature*, its *position*, and its *shape* in terms of the covariance matrix of the C$\alpha$ coordinates in the region. We argue that this representation of protein spatial layouts finds a favorable tradeoff between a single global annotation, such as a text prompt or protein family, and more complex shape descriptors, such as meshes or voxel grids. A global annotation may be insufficient to provide the desired control over the spatial layout, and a more complex annotation could be difficult to generate or specify without training an additional model. Meanwhile, 3D ellipsoids are expressive and precise, yet easy to generate and manipulate.

Mathematically, we define a protein spatial layout consisting of $K$ *ellipsoids* as an unordered set $\mathbf{E} = \{E_k = (\mu_k, \Sigma_k, f_k, n_k)\}_{k \in \{1...K\}}$ where each ellipsoid is represented as a Gaussian with mean $\mu_k \in \mathbb{R}^3$, covariance $\Sigma_i \in \mathbb{R}^{3\times3}$, count $n_k \in \mathbb{N}^+$, and feature annotation $f_k \in \mathcal{X}$ where $\mathcal{X}$ is the application-dependent feature space. Viewed as Gaussian probability distributions, our ellipsoids do not have well-defined boundaries; however, for visualization and evaluation purposes, we define the ellipsoid boundary to be the surface at Mahalanobis distance $\sqrt{5}$, i.e.,

$$\partial E_k = \left\{ x \in \mathbb{R}^3 : \sqrt{(x - \mu_k)^T \Sigma_k^{-1} (x - \mu_k)} = \sqrt{5} \right\} \quad (1)$$

This is the functional form of a conventional ellipsoid. The distance $\sqrt{5}$ is chosen so that $83\%$ of the density falls inside the surface, which yields the best visual results (Appendix Fig 12).

**Ellipsoid Segmention.** Provided a protein structure, obtaining its ellipsoid representation (e.g., for training purposes) consists of two steps: *segmentation* of the protein into semantically coherent regions, and *extraction* of ellipsoid descriptions $(\mu_k, \Sigma_k, n_k, f_k)$ for each region. To segment the protein, we consider a simple, non-learned segmentation algorithm that places two residues in the same region if and only if they are both *spatially* proximal and *semantically* similar. We construct a segmentation graph by drawing an edge for each such pair of residues and return the list of connected components of this segmentation graph. For each segmented region, we aggre-

gate the residue features to obtain $f_k$ and compute the mean and covariance of the C$\alpha$ positions. These steps are illustrated in Figure 2 and further detailed in Appendix Algorithm 3. We found this to be more reliable than more sophisticated variants using, e.g., $K$-means or spectral clustering.

In this work, we focus on 3D ellipsoids specifying *secondary structure* layouts, i.e., regions of $\alpha$-helices and $\beta$-sheets. Our feature space is thus a two-class space of secondary structures types, $f_k \in \mathcal{X} = \{\alpha, \beta\}$. We featurize residues using DSSP (Kabsch & Sander, 1983) and draw edges in the segmentation graph between amino acids with the same secondary structure label and within 5 Å. The ellipsoid annotation $f_k$ then simply inherits the label of its constituent residues. We exclude all loop residues and ellipsoids with five or fewer residues.

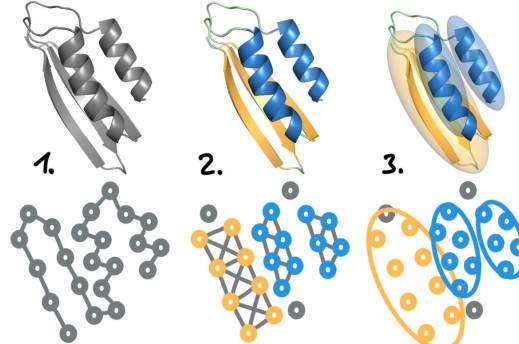

Figure 2: **Extracting ellipsoids from data.** *1)* Input protein. *2)* Annotate residues and draw edges if they have the same feature AND are within 5Å. *3)* Fit Gaussians to connected components.

## 3.2 ELLIPSOID CONDITIONING

**Unconditional model.** We inject our ellipsoid conditioning into an existing protein structure generative model for which we choose Multiflow (Campbell et al., 2024), which jointly generates sequence and structure. Following their framework, we generate proteins represented as an array of *frames* $\boldsymbol{T} \in SE(3)^N$ where each residue's frame $\boldsymbol{T}_i = (R_i, \mathbf{t}_i) \in SE(3)$ has an associated translation $\mathbf{t}_i \in \mathbb{R}^3$ and rotation matrix $R_i \in \mathbb{R}^{3\times 3}$ constructed from backbone coordinates following AlphaFold2 (Jumper et al., 2021). Additionally, each residue has an amino acid type $\mathbf{a}_i \in \{1 \dots 20\}$.

To jointly generate the translations, rotations, and amino acids, Multiflow employs three types of flow matching procedures that iteratively update all three modalities. Translations are handled with linear flow matching from a Gaussian prior (Lipman et al., 2022), rotations with Riemannian flow matching on $SO(3)$ (Chen & Lipman, 2024), and residue types with discrete flow matching (Campbell et al., 2024; Gat et al., 2024), resulting in a joint flow that transports from a prior $p_0(\mathbf{t}, R, \mathbf{a})$ to the data distribution $p_1(\mathbf{t}, R, \mathbf{a})$ while tracing out a probability path $p_t(\mathbf{t}, R, \mathbf{a})$ where $t \in [0, 1]$.

The flow is parameterized by a single backbone architecture with translations, rotations and residue type inputs from which it predicts a time dependent translation vector field $v_{\theta,t}^{\text{tr}}(\mathbf{t})$, rotation vector field $v_{\theta,t}^{\text{rot}}(R)$, and a rate matrix $\mathcal{R}_{\theta,t}(\mathbf{a})$ dictating residue type updates. The architecture is composed of several identical *update blocks*, each of which updates $d$-dimensional residue representations $\mathbf{s}_i \in \mathbb{R}^d$ for $i \in \{1, \dots N\}$, residue pair representations $\mathbf{z}_{i,j} \in \mathbb{R}^d$ for $i, j \in \{1 \dots N\}$ and residue frames $\mathbf{T}_i$ for $i \in \{1, \dots N\}$. The updates are $SE(3)$-equivariant and accomplished with a mixture of shallow transformers and Invariant Point Attention (Jumper et al., 2021); we refer to Campbell et al. (2024) for complete architectural details. After all the update blocks, the final residue tokens $\mathbf{s}_i$ and frames $\mathbf{T}_i$ are used to parameterize the flow fields $v_{\theta,t}^{\text{tr}}(\mathbf{t})$, $v_{\theta,t}^{\text{rot}}(R)$, $\mathcal{R}_{\theta,t}(\mathbf{a})$

**Injecting Ellipsoid Conditioning.** We now aim to fine-tune a pre-trained unconditional Multiflow model trained to sample $p_1(\mathbf{t}, R, \mathbf{a})$ toward sampling an ellipsoid conditioned density $p_1(\mathbf{t}, R, \mathbf{a} \mid \mathbf{E})$ and obtain ProtComposer. At inference time, ellipsoids can be specified manually or sampled from a second distribution $p(\mathbf{E})$ of novel and diverse ellipsoids (see Section 3.4) to target the density $p_1(\mathbf{t}, R, \mathbf{a} \mid \mathbf{E})p(\mathbf{E})$. For fine-tuning, we only provide the conditioning information as additional input - the training loss remains the unchanged loss of Multiflow. To inject the ellipsoid information, we follow best principles from semantic map conditioning for image diffusion models (Zhang et al., 2023b; Li et al., 2023b) and design architecture modifications that minimally perturb the unconditional model at the time of initialization. That is, with an empty set of ellipsoids as input, the untrained *conditional* model should produce identical outputs as the unconditional model. This is accomplished by preserving the initial residue representations $\mathbf{s}_i, \mathbf{z}_{ij}, \mathbf{T}_i$, and only supplying additional information from 3D ellipsoids to inform their updates, described below.

We introduce additional tokens $\mathbf{e}_k \in \mathbb{R}^d$ for each ellipsoid $k \in \{1 \dots K\}$ that are of the same dimensionality $d$ as the residue tokens $\mathbf{s}_i$. These tokens are initialized with embeddings of all $SE(3)$-invariant quantities of ellipsoids—their size $n_k$, squared radius of gyration $\text{tr}\,\Sigma$, and sec-

| **Algorithm 1:** Invariant Cross Attention | **Algorithm 2:** Update Block |
|---|---|
| **Input:** Residue tokens $\mathbf{s}_i$ and frames $\mathbf{T}_i = (R_i, \mathbf{t}_i)$; ellipsoid parameters $\mathbf{E}_k = (\mu_k, \Sigma_k)$ | **Input:** Residue tokens $\mathbf{s}_i$, pair reps $\mathbf{z}_{ij}$, residue frames $\mathbf{T}_i$, ellipsoid tokens $\mathbf{e}_k$, ellipsoid parameters $\mathbf{E}_k = (\mu_k, \Sigma_k)$ |
| $\mathbf{r}_{ik} \leftarrow T_i^{-1} \circ \mu_k$ | $\mathbf{s}\ \text{+=}\ \text{InvariantPointAttention}(\mathbf{s}, \mathbf{z}, \mathbf{T})$ |
| $C_{ik} \leftarrow R_i \Sigma_k R_i^T$ | $\mathbf{s}\ \text{+=}\ \text{InvariantCrossAttention}(\mathbf{s}, \mathbf{T}, \mathbf{E})$ |
| $\mathbf{a}_{ik} = \mathbf{s}_i + \text{Linear}(\text{PosEmbed}(\mathbf{r}_{ik}))$ | $\mathbf{s} \leftarrow \text{Concat}(\mathbf{s}, \mathbf{e})$ |
| $\mathbf{a}\ \text{+=}\ \text{Linear}(\text{Flatten}(C_{ik}))$ | $\mathbf{s}\ \text{+=}\ \text{Transformer}(\mathbf{s})$ |
| $\mathbf{q}_i = \text{Linear}(\mathbf{s}_i)$ | $\mathbf{s}, \mathbf{e} \leftarrow \text{Split}(\mathbf{s})$ |
| $\mathbf{k}_{ik}, \mathbf{v}_{ik} = \text{Linear}(\mathbf{a}_{ik})$ | $\mathbf{T} \leftarrow \text{RigidUpdate}(\mathbf{s}, \mathbf{T})$ |
| $\mathbf{s}_i\ \text{+=}\ \text{Attention}_k(\mathbf{q}_i, \mathbf{k}_{ik}, \mathbf{v}_{ik})$ | $\mathbf{z}\ \text{+=}\ \text{EdgeUpdate}(\mathbf{s})$ |

ondary structure type $f_k$. Then, in each model layer, these tokens inform the updates of the residue representations $\mathbf{s}_i, \mathbf{z}_{ij}, \mathbf{T}_i$ (and are themselves updated) via two mechanisms:

- To update the residue tokens $\mathbf{s}_i$ with information about the location and shapes of the ellipsoids, we introduce a novel *invariant cross attention* mechanism (Algorithm 1 and Figure 1) whereby values are aggregated from the ellipsoid tokens in an $SE(3)$-invariant manner. Similar to IPA, this mechanism uses the residue local frames to enforce invariance, although the ellipsoid tokens are not themselves updated, which we discuss further in Appendix B.

- To provide a mechanism for residue and ellipsoid tokens to mutually update each other, we concatenate the tokens along the sequence dimension right before the Transformer stack, and re-split the sequence afterwards.

All other aspects of the Multiflow update blocks, such as the frame update and edge update layers, remain architecturally unmodified. In Algorithms 1 and 2, we outline the new invariant cross attention alongside the modified update block with modifications colored green.

## 3.3 GUIDANCE FOR THE SELF-CONDITIONED AND JOINT FLOW

After fine-tuning a base protein structure generative model that samples $p_\theta(\mathbf{t}, R, \mathbf{a}) \approx p_1(\mathbf{t}, R, \mathbf{a})$ to obtain ProtComposer's distribution $p_\theta(\mathbf{t}, R, \mathbf{a} \mid \mathbf{E})$, we interpolate between the two distributions via *classifier-free guidance* (Ho & Salimans, 2022) controlled by a guidance parameter $\lambda \geq 0$. This enables finding the optimal $\lambda$ to trade off between the designability of $p_\theta(\mathbf{t}, R, \mathbf{a})$ that is recovered with $\lambda = 0$ and the diversity, novelty, and ellipsoid adherence of $p_\theta(\mathbf{t}, R, \mathbf{a} \mid \mathbf{E})$ corresponding to $\lambda = 1$. Interpolations for individual samples are visualized in Figure 3.

Implementing such guidance is complicated by the facts that we model the flow field instead of the score (as in diffusion models), that ProtComposer's conditional probability paths are not Gaussian (the guided flows of (Ho & Salimans, 2022) are not directly applicable), and since we employ *self-conditioning*. Before elaborating on the self-conditioning difficulty, we lay out how we guide the joint flow over translations, rotations, and discrete residue types by separately interpolating their flow fields at each inference step:

- Translations: we interpolate the unconditional vector field $v_{\theta,t}^{tr}(\mathbf{t})$ and the conditioned version $v_{\theta,t}^{tr}(\mathbf{t}, \mathbf{E})$ as $\lambda v_{\theta,t}^{tr}(\mathbf{t}, \mathbf{E}) + (1 - \lambda)v_{\theta,t}^{tr}(\mathbf{t})$. Since the conditional probability paths for translations are Gaussian paths, this corresponds to the guided flows of Zheng et al. (2023b), which sample the same approximation of the unconditional distribution tilted by the conditional distribution as guided diffusion models, i.e., an approximation of $p_1(\mathbf{t})^{(1-\lambda)}p_1(\mathbf{t} \mid \mathbf{E})^\lambda$ if we were to interpolate models that only sample translations.

- Rotations: while with our non-Gaussian paths on SO(3) the results of Zheng et al. (2023b) no longer hold, we find empirical success in using, in analogy to translations, $\lambda v_{\theta,t}^{rot}(\mathbf{t}, \mathbf{E}) + (1 - \lambda)v_{\theta,t}^{rot}(\mathbf{t})$, which follows Yim et al. (2024b).

- Discrete Flow: we construct the rate matrix for the discrete flow as the expectation of the conditional rate matrix (see Campbell et al. (2024)) over predicted probabilities of the denoised residues that we obtain as a combination of the unconditional model's predictions and the ellipsoid conditioned model's predictions. Specifically, we use the unconditionally predicted probabilities tilted by the ellipsoid conditioned probabilities $p_\theta(\mathbf{a}_i^{(1)} \mid \mathbf{a}_i^{(t)})^{(1-\lambda)}p_\theta(\mathbf{a}_i^{(1)} \mid \mathbf{a}_i^{(t)}, \mathbf{E})^\lambda$, where the superscript denotes denoising time.

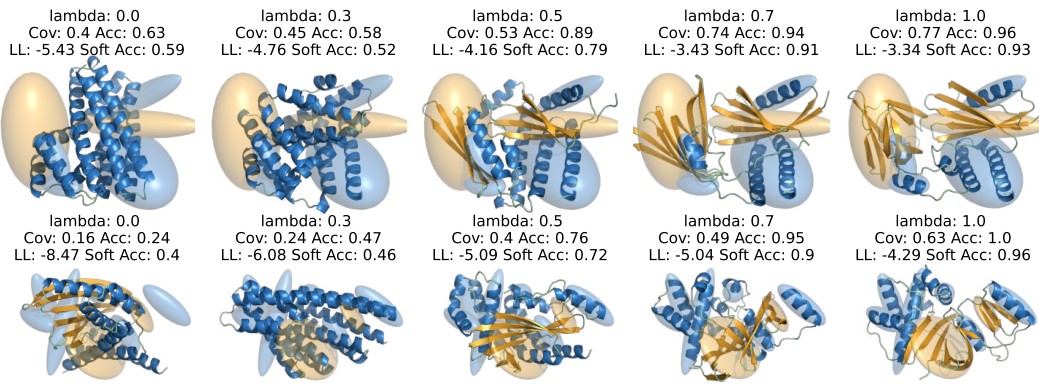

Figure 3: Two protein layouts (top and bottom) and generations for them with varying guidance strength; no guidance ($\lambda = 0$) on the left, full guidance ($\lambda = 1$) on the right. Ellipsoid alignment metrics are labeled as *Cov*=coverage, *Acc*=accuracy, *LL*=likelihood, and *Soft Acc*=soft accuracy.

Both Multiflow and ProtComposer use *self-conditioning* (Chen et al., 2023), in which, during inference, the flow-model receives the output of the previous integration step as additional *self-conditioning input*. During inference, the unconditional model $p_\theta(\mathbf{t}, R, \mathbf{a})$ produces the self-conditioning variable $X$, and from the ellipsoid conditioned model $p_\theta(\mathbf{t}, R, \mathbf{a} \mid \mathbf{E})$, we obtain $X_\mathbf{E}$. Instead of supplying $X$ to the unconditional and $X_\mathbf{E}$ to the conditioned model, we use $\lambda X_\mathbf{E} + (1-\lambda)X$ for both, which achieves better designability and ellipsoid adherence for all $\lambda$. An exploration and ablation of self-conditioning variants is in Appendix C.2.

### 3.4 GENERATING NOVEL ELLIPSOIDS

In designing the ellipsoid conditioning mechanism, we have so far made no assumptions about the *sources* of ellipsoids provided during inference. Next to using *manually specified* ellipsoids, there is also an opportunity in sampling *synthetic ellipsoids* from an additional generative model $p_\theta(\mathbf{E})$ to sample an unconditional distribution of protein structures factorized as $p_\theta(\mathbf{t}, R, \mathbf{a} \mid \mathbf{E})p_\theta(\mathbf{E})$. While it may be tempting to use a deep learning solution for $p_\theta(\mathbf{E})$, we purposefully avoid this and argue that the factorization is best leveraged with a simple statistical model over ellipsoid layouts. Instead of a deep learned $p_\theta(\mathbf{E})$ that may produce layouts that are similar to the training data, a simple statistical model for $p_\theta(\mathbf{E})$ guarantees sampling diverse and *novel* layouts, which lead to more diverse and novel protein structures from $p_\theta(\mathbf{t}, R, \mathbf{a} \mid \mathbf{E})p_\theta(\mathbf{E})$ - properties that are crucial for protein design where the aim is commonly to produce novel designs.

To generate novel ellipsoid layouts, we first sample means and covariances for $K$ ellipsoids and then assign secondary structure and residue count annotations. The model over means and covariances is

$$p\left(\{(\mu_k, \Sigma_k)\}_{k=1}^K\right) \propto \left[\prod_{k=1}^K \mathcal{N}\left(\mu_k, \mathbf{0}, \sigma^2 \mathbf{I}_3\right) \mathcal{W}_3(\Sigma_k; \psi^2 \mathbf{I}_3, \nu)\right] \exp\left(-U(\{(\mu_k, \Sigma_k)\}_{k=1}^K)\right), \quad (2)$$

$$U(\{(\mu_k, \Sigma_k)\}_{k=1}^K) = \sum_{k \neq j} \frac{1}{\left[(\mu_k - \mu_j)^T \Sigma_k^{-1} (\mu_k - \mu_j)\right]^2}. \quad (3)$$

That is, the ellipsoid means and covariances are drawn i.i.d. from isotropic Gaussian and Wishart distributions, respectively, and multiplied with the Boltzmann factor of an energy function that penalizes ellipsoid overlaps. Intuitively, $\sigma$ controls the ellipsoid's *spread*, $\psi$ controls their *volume*, $\nu$ controls their anisotropy or *"roundness"*, and $U$ prevents overlaps. The energy $U$ is a simple inverse square repulsion based on pairwise Mahalanobis distances. We sample this via rejection sampling, i.e., by sampling $\mu_i, \Sigma_i$, evaluating their energy $U$, and rejecting with probability $e^{-U}$.

To choose the ellipsoid annotations, we first independently annotate each ellipsoid as $\alpha$ with probability $\gamma$ and $\beta$ with probability $1 - \gamma$. We then observe that for a given choice of $\{\alpha, \beta\}$, the ellipsoid volume $\sqrt{\det \Sigma_i}$ strongly determines the residue count by a simple linear fit (Appendix 10). Hence, we use this linear fit to assign the number of residues instead of modeling it independently.

Table 1: **Recapitulating layouts of PDB proteins.** Protein layout specification adherence metrics for ProtComposer conditioned on layouts of validation set proteins under different levels of guidance strength $\lambda$. Multiflow as random baseline. The *Oracle* metrics are computed by treating the validation set proteins, which the layouts were drawn from, as generated proteins.

| Model | Geometric | | | Probabilistic | | |
|---|---|---|---|---|---|---|
| | Coverage ↑ | Misplacement ↓ | Accuracy ↑ | Likelihood ↑ | Soft Accuracy ↑ | JSD ↓ |
| Multiflow | 0.49 | 0.40 | 0.59 | -5.0 | 0.61 | 0.43 |
| Chroma | 0.59 | 0.36 | 0.58 | -4.2 | 0.58 | 0.48 |
| ProtComposer $\lambda=0.2$ | 0.55 | 0.33 | 0.66 | -4.5 | 0.65 | 0.36 |
| ProtComposer $\lambda=0.4$ | 0.62 | 0.28 | 0.75 | -4.0 | 0.73 | 0.28 |
| ProtComposer $\lambda=0.6$ | 0.70 | 0.21 | 0.85 | -3.5 | 0.83 | 0.20 |
| ProtComposer $\lambda=0.8$ | 0.75 | 0.18 | 0.90 | -3.3 | 0.88 | 0.15 |
| ProtComposer $\lambda=1.0$ | 0.78 | 0.17 | 0.90 | -3.2 | 0.90 | **0.13** |
| ProtComposer $\lambda=1.6$ | **0.86** | **0.13** | **0.93** | **-3.0** | 0.91 | **0.13** |
| ProtComposer $\lambda=2.0$ | 0.79 | 0.14 | 0.92 | -3.2 | **0.92** | **0.13** |
| Oracle | 0.89 | 0.07 | 0.92 | -2.8 | 0.93 | 0.15 |

## 4 EXPERIMENTS

Starting from the publicly available pre-trained checkpoint, we fine-tune Multiflow (Campbell et al., 2024) on the dataset and splits supplied by the authors, where ellipsoid spatial layouts are obtained for each protein as described in Section 3.1. We train only on the joint unconditional modeling task (i.e., no motif scaffolding, inverse folding, or forward folding). At inference time, we employ self-conditioning, rotational annealing, and 500 inference steps as described in Campbell et al. (2024). For guidance, we use the pretrained Multiflow checkpoint as the unconditional model.

Throughout our experiments, we consider three sources of ellipsoid layouts: PDB proteins from the Multiflow validation set (*data ellipsoids*), ellipsoids drawn from our statistical model (Section 3.4; *synthetic ellipsoids*), and *manually specified ellipsoids*. The key feature of data ellipsoids is that they are associated with ground-truth proteins, providing an oracle generator for ellipsoid adherence. When using data ellipsoids, we sample proteins of equal lengths to the ground-truth proteins, while for novel ellipsoids, the protein length is the sum of ellipsoid residue counts, $\sum_k n_k$. Summary statistics about both sources of ellipsoids are described in Appendix A.

### 4.1 ELLIPSOID CONSISTENCY

Given a 3D ellipsoid layout and a protein generated based on the layout, we seek to quantify the degree of consistency and alignment between the protein and the layout. We define two classes of metrics: three *geometric* metrics, in which we interpret ellipsoids as ellipsoids with a definite interior and exterior based on Mahalanobis distance (Eq. 1), and three *probabilistic* metrics, in which we only use the Gaussian parameters associated with the ellipsoid. Layouts are provided in a fixed orientation, and the model equivariantly generates proteins in the same orientation, so no roto-translational alignment is performed for any of these metrics:

- **Coverage** ↑—the fraction of structured residues located inside at least one ellipsoid.
- **Misplacement** ↓—the sum of errors $\sum_k |p_k - p'_k|$ between $p_k := n_k / \sum_{k'} n_{k'}$, the residue fraction of ellipsoid $k$, and $p'_k$, the actual fraction of structured residues inside ellipsoid $k$
- **Accuracy** ↑—the fraction of residues located inside at least one ellipsoid with the same secondary structure type as the ellipsoid annotation (residues can be counted multiple times).
- **Likelihood** ↑—we view the ellipsoid layout as a Gaussian mixture model (GMM), whose density is renormalized to integrate to $\sum_k n_k$. We then report the average log density of each structured residue position under this renormalized GMM.
- **Soft Accuracy** ↑—given an ellipsoid layout viewed as a GMM, we can compute a posterior distribution over secondary structure type $f \in \{\alpha, \beta\}$ for a residue located at arbitrary position $\mathbf{x}$ in 3D space, given by $p(f \mid \mathbf{x}) \propto p(f, \mathbf{x}) \propto \sum_{k: f_k = f} n_k \mathcal{N}(\mathbf{x}; \mu_k, \Sigma_k)$. We

then report the mean (over structured residues) of the normalized probability assigned to the actual secondary structure type of the generated residues.

- **Resegment JSD** ↓—we recompute the ellipsoid layout from the generated protein structure and view both ellipsoid layouts as defining Gaussian mixture models over the event space $\{\alpha, \beta\} \times \mathbb{R}^3$. We then report the Jensen-Shannon divergence between the two distributions.

Note that Coverage, Misplacement, and Likelihood only quantify the alignment between the overall *shape* of the ellipsoid layout and the protein, whereas Accuracy, Soft Accuracy, and Resegment JSD also take into consideration the secondary structure annotations.

In Table 1, we generate proteins from layouts specified by ellipsoids drawn from the validation set of PDB proteins and we report the average for all metrics. We sample for various levels of guidance $\lambda$ with ProtComposer, where $\lambda = 1$ corresponds to the purely conditional model. We compare with Chroma (Ingraham et al., 2023) conditioned on ellipsoid layouts via its inference time `ShapeConditioner` functionality (details in Appendix A.2). For oracle and random baselines, we compute the ellipsoid alignment of the ground truth protein and of a protein generated from pre-trained Multiflow without any ellipsoid conditioning, respectively. We observe that ellipsoid alignment sharply increases when the guidance strength is increased above $\lambda = 0.5$ and quickly approaches the level of alignment of the oracle. Hence, 3D ellipsoids provide highly effective control over protein layouts. Figure 3 visualizes examples of protein generations and their associated alignment metrics for various guidance levels. More examples in Appendix Figure 24.

## 4.2 IMPROVED DIVERSITY AND NOVELTY

We now show that conditioning on ellipsoid layouts can improve the diversity (and related metrics) of Multiflow generations. Without conditioning, Multiflow and related methods generate highly designable proteins, but exhibit limited secondary structure diversity (73% helices instead of the 42% of PDB proteins) and low complexity by visual inspection (Figure 5). These effects can be ameliorated by decreasing the rotational annealing, but this results in rapidly deteriorating designability (Figure 4). We thus construct a pipeline in which *synthetic ellipsoids* are drawn from our statistical model and provided as conditioning input to ProtComposer. Conceptually, this can be thought of as manually controlling the "ellipsoid marginal" of the generated distribution, where the ellipsoids amount to incomplete, compressed observations of the full state space (i.e., backbone structure).

To explore this pipeline systematically, we first fix $K = 5$ ellipsoids and $\psi = 5$ Å in our ellipsoid statistical model, a setting that consistently produces proteins of length 120–200. We then sweep over all combinations of protein compactness $\sigma \in [3, 4 \ldots 10]$, helix fraction $\gamma \in [0.2, 0.4, 0.6, 0.8, 1]$ and ellipsoid anisotropy $\nu \in [5, 10, 20, 50, 100]$, and guidance strength $\lambda \in [0.1, 0.2 \ldots 1.0]$. For each setting, we draw 100 synthetic ellipsoid layouts, generate a protein for each ellipsoid, and evaluate the set of 100 proteins on the following metrics:

- **Designability** ↑—the fraction of structures for which at least one out of 8 sequences sampled by ProteinMPNN (Dauparas et al., 2022) results in scRMSD $< 2\mathring{A}$ when re-folded with ESMFold (Lin et al., 2023).
- **Diversity** ↑—the Vendi score (Friedman & Dieng, 2023) of the set of structures with TM-Score (Zhang & Skolnick, 2005) as the similarity kernel, ranging from 0 to 100.
- **Novelty** ↑—one minus the average TM-Score to the closest chain in the PDB as retrieved by FoldSeek (Van Kempen et al., 2024).
- **Helicity** ↓—the fraction of structured residues which are assigned $\alpha$-helix by DSSP. While helical proteins are not undesirable *per se*, we wish to increase the secondary structure diversity of Multiflow proteins to be more similar to naturally observed helicity.

In Figure 4, we show the performance of all 1750 inference settings, with Pareto frontiers shown for the tradeoff between designability and each of the three other metrics. We compare with Multiflow's frontier from varying its rotational inference annealing parameter (elaborated in Appendix A.4) which acts analogous to diffusion model low-temperature sampling to trade off diversity for fidelity (Yim et al., 2023a; Bose et al., 2024). Further, we sweep Multiflow across protein length distributions that match the length distributions of our ellipsoid statistical model (details in Appendix A). For Chroma (Ingraham et al., 2023) and RFDiffusion (Watson et al., 2023), we sweep over the

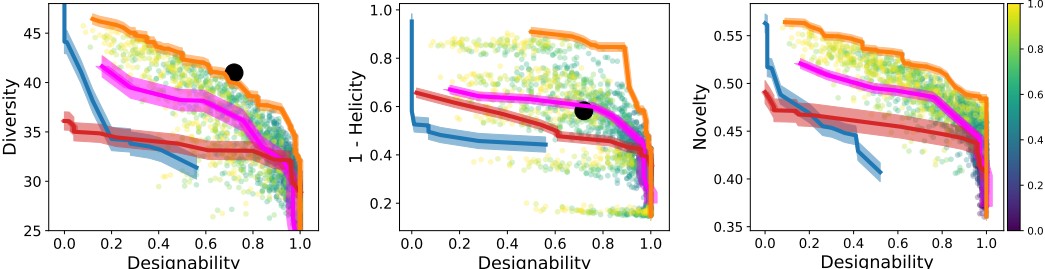

Figure 4: **ProtComposer** Pareto frontiers traced out by our ellipsoid statistical model's family of distributions and by varying guidance strength. **Multiflow**—varying rotational annealing strength. **Chroma** and **RFDiffusion**—varying sampling temperature. **Black dot •:** metrics of PDB proteins. Each colored point (by guidance strength) is one ProtComposer distribution. Shaded areas are standard deviations from 4 seeds for points on the frontiers (App. A).

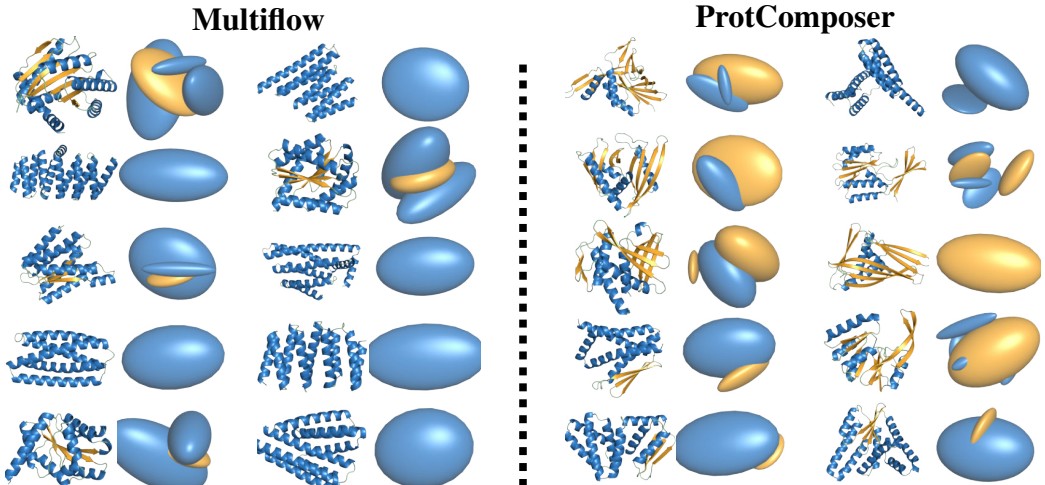

Figure 5: **Random samples** with extracted ellipsoids at segmentation threshold 6.5Å from Multiflow (left) and our synthetic ellipsoid pipeline with $\sigma = 10$ Å , $\nu = 100$, $\gamma = 0.5$ and $\lambda = 0.6$ (right). 6 out of 10 Multiflow proteins degenerate to a single $\alpha$-helix ellipsoid, while ProtComposer's proteins exhibit better secondary structure diversity and ellipsoid compositionality.

same length distributions and their sampling temperature. We robustly observe that ellipsoid conditioning enables the generation of more diverse, more novel proteins with a helicity that is closer to PDB proteins while retaining a higher level of designability than previously possible with Multiflow. Appendix Figure 18 visualizes examples of this expanded Pareto front, and Appendix Figure 23 shows scatterplots to examine the impacts of varying $\sigma, \nu, \gamma$, and $\lambda$.

By increasing diversity and related metrics, ellipsoid conditioning also improves the aggregate similarity of generated proteins to proteins from the PDB. In Table 2, we quantitatively ascertain that proteins from pre-trained Multiflow without ellipsoid conditioning are more helical, less diverse, and less compositional than PDB proteins. For this, we quantify a protein's **compositionality** by splitting it into components of residues that are close to each other (6.5Å as in Figure 5) and of equal secondary structure. Then, with $m_k$ as the count of residues in the $k$-th component, we report the *effective number of components* $\exp(-\sum_k p_k \log p_k)$ where $p_k = m_k / \sum_k m_k$ can be interpreted as a residue's probability for the $k$-th component. Table 2 shows that, by generating proteins

Table 2: **Recovering statistics of PDB proteins.** The diversity, helicity, and compositionality of PDB proteins compared with Multiflow and ProtComposer.

| Model | Div. | Helic. | Comp. |
|---|---|---|---|
| Multiflow | 29 | 73% | 1.9 |
| Ours ($\lambda = 0.2$) | 32 | 72% | 1.8 |
| Ours ($\lambda = 0.4$) | 36 | 68% | 2.1 |
| Ours ($\lambda = 0.6$) | 38 | 57% | 2.6 |
| Ours ($\lambda = 0.8$) | 40 | 52% | 2.8 |
| Ours ($\lambda = 1.0$) | **41** | 51% | 2.9 |
| Ours ($\lambda = 1.6$) | 45 | 48% | 3.1 |
| Ours ($\lambda = 2.0$) | 48 | **47%** | **3.2** |
| PDB proteins | 41 | 42% | 4.1 |

based on *data ellipsoids* (from the Multiflow validation set), ProtComposer largely closes the gap to PDB proteins and restores the aggregate levels of helicity, diversity, and compositionality.

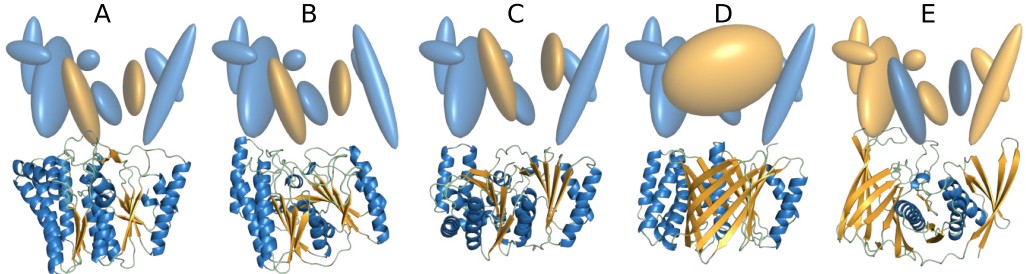

Figure 6: **Controlled manipulation of ellipsoid layouts** from PDB ID 8F9Y_A. (A) Original ellipsoid layout and generated protein. (B) Rotating the rightmost $\alpha$-helix. (C) Translating the $\beta$-sheets upwards. (D) Merging and expanding the $\beta$-sheet regions. (E) Secondary structure inversion.

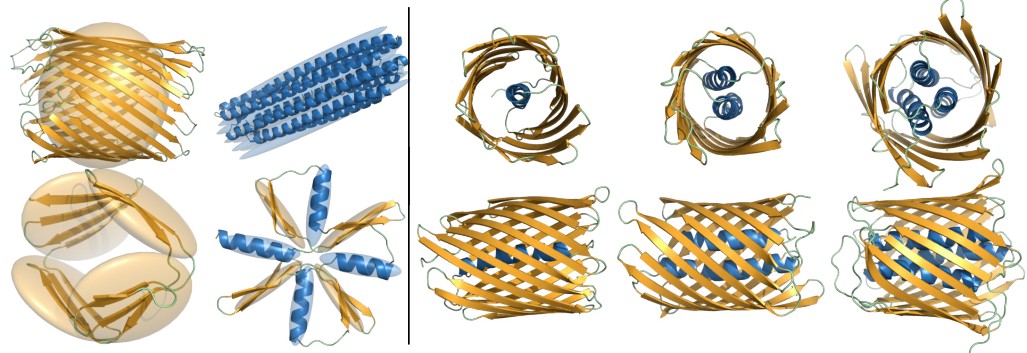

Figure 7: **Proteins generated from hand-constructed ellipsoids.** *Left:* 4 protein-layouts with generated proteins. *Right:* 3 beta-barrels containing varying numbers of helices generated from a large beta-sheet ellipsoid containing elongated helix ellipsoids (both top and side views shown).

### 4.3 FLEXIBLE CONDITIONING

To fully realize and demonstrate the potential of ProtComposer, we explore manual construction or manipulation of ellipsoid layouts in various settings. First, in Figure 6 we demonstrate that by fixing the noise and changing the ellipsoid parameters, we can impose fine-grained control and manipulation of generated proteins. For example, we can move individual secondary structure elements, enlarge or merge them, or convert between them. When applied to PDB protein layouts, this capability enables precise *structural editing* of existing proteins, an exciting and first-in-class capability. Next, in Figure 7, we test the boundaries of the model's generalization ability by manually constructing ellipsoids in eclectic layouts. These result in, for example, extremely long helix bundles, massive $\beta$-barrels or $\beta$-sheets, and peculiar arrangements of these artificial elements. Although these extreme structures are not always designable, they demonstrate the creative and faithful alignment of the fine-tuned model to the provided ellipsoid conditioning.

## 5 CONCLUSION

We developed ProtComposer for conditioning protein structure generative models on 3D ellipsoid layouts. We implemented it for Multiflow and secondary structure annotated ellipsoids, using novel architectural components such as *Invariant Cross Attention*. We quantitatively ascertained that generations tightly adhere to the input ellipsoid layouts and provided a range of examples to demonstrate how this enables reliably generating proteins with desired layouts via hand-specified ellipsoids and editing existing proteins. To generate unconditionally, we condition on novel and diverse ellipsoid layouts drawn from a newly developed statistical model. This produced a family of protein structure distributions that far surpasses the Pareto frontiers of designability to novelty and diversity achieved by Multiflow. ProtComposer generates more compositional proteins with a helicity close to that of PDB proteins while maintaining designability. We anticipate expanding the annotation types of our ellipsoids to function specifications to further push the frontier of controllable protein design.

## 6 ACKNOWLEDGEMENTS

This work was primarily developed during Bowen Jing's and Hannes Stark's internships at NVIDIA. Tommi Jaakkola and Jason Yim acknowledge support and Hannes Stark and Bowen Jing acknowledge partial support from the Machine Learning for Pharmaceutical Discovery and Synthesis (MLPDS) consortium, the DTRA Discovery of Medical Countermeasures Against New and Emerging (DOMANE) threats program, and the NSF Expeditions grant (award 1918839) Understanding the World Through Code.

We thank Gabriele Corso, Felix Faltings, Rachel Wu, and Weili Nie for helpful discussions and feedback.

## 7 REPRODUCIBILITY STATEMENT

Our implementations are based on the architecture, training losses, optimizers, datasets, and data processing of Multiflow (Campbell et al., 2024) for which the authors provide code at `https://github.com/jasonkyuyim/multiflow`. All code for this paper is available at `https://github.com/NVlabs/protcomposer`, and we provide the following descriptions to enable reproducing results based on the manuscript alone: We provide algorithm 1 to specify our invariant cross-attention mechanism, which can be implemented with invariant point attention classes at `https://github.com/aqlaboratory/openfold` as a starting point. The modifications of the Multiflow update block are specified in Algorithm 2. Our ellipsoid extraction procedure is detailed in algorithm 3. All parameters for sweeps to produce the Pareto frontiers in Figure 4 are specified in Section 4.2 and our Appendix on Pareto frontier details A.4. We provide details on how we run Multiflow as a baseline in Appendix A.4. Details on running Chroma and how we implement ellipsoid conditioning for Chroma are available in Appendix A.2. Training details such as duration and resource requirements are in Appendix A.1. We describe the full form of our Ellipsoid statistical model and how to sample it in Section 3.4. All our metrics are specified explicitly in Sections 4.2 and 4.1 with citations to tools required for their computation.

## 8 ETHICS STATEMENT

We present a general method for protein structure generation that may be used to aid protein design. Proteins are versatile tools that carry out biological functions. Depending on the purpose our tool is used for—depending on which protein design it is employed to aid—its use is either ethical or unethical. The vast majority of current use cases of similar technology are with the intent of achieving positive outcomes for humanity via, e.g., developing drugs, biomolecules that accelerate industrial processes, or proteins used as tools to improve our understanding of biology itself. Thus, we estimate that the ethical use cases with potential for positive impact on humanity will far outweigh any negative impacts from unethical uses.

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

## A    EXPERIMENTAL DETAILS

### A.1    IMPLEMENTATION DETAILS

---

**Algorithm 3:** Residue Segmentation and Gaussian Fitting

---

**Input:** Positions of alpha carbons **pos**, Secondary structure annotation **ss**, Radius threshold
     **threshold**
**Output:** List of ellipsoids with secondary structure type, position, and covariance.
**distmat** $\leftarrow$ **pairwise_distances(pos,pos)**
**edges** $\leftarrow$ **argwhere**$((\textbf{distmat} < \textbf{threshold}) and (\textbf{ss}_{[None]} == \textbf{ss}_{[:,None]}))$
$G \leftarrow$ **graph_from_edges(edges)**
**components** $\leftarrow$ **extract_connected_components(G)**
**ellipsoids** $\leftarrow$ [ ]
**for** *component* in *components* **do**
    **component** $\leftarrow$ list(**component**)
    **if** *ss[component[0]]* $==$ *loop* **then**
        ⌊ **continue**
    **if** *len(component)* $< 5$ **then**
        ⌊ **continue**
    **ellipsoids.append**({
    'type': **ss[component**[0]],
    'position': **pos[component].mean()**
    'covariance': **covariance(pos[component]**.T)
    })
**return** *ellipsoids*

---

**Training.** We finetune Multiflow (Campbell et al., 2024) starting from a checkpoint provided on the authors' GitHub and use their optimizers, data filtering, losses, and hyperparameters (AdamW, learning rate 0.0001). To monitor training progress, we run inference conditioned on data ellipsoids and report designability and ellipsoid adherence metrics. We do not employ early stopping based on any of the metrics. Training is carried out on 8 NVIDIA A100 GPUs for 20 hours, corresponding to 83 epochs.

**Data.** The training data consists of PDB proteins and synthetic data. The PDB training colleted by Yim et al. (2023b) consists of 18684 proteins of length 60-384. We train on crops of size 256. Additional training data are 4179 synthetically generated proteins of Multiflow with high designability (see Campbell et al. (2024)). The ellipsoids to condition on at test time are extracted from the PDB proteins and synthetic proteins as described in Section 3.1, Figure 2 and detailed in Algorithm 3. This means that, at training time, the number of ellipsoids is determined by the data. In Figure 11, we provide a histogram of the number of ellipsoids per protein in PDB proteins. We trained additional models (no results included in the paper) on the AlphaFold2 (Jumper et al., 2021) database following Lin et al. (2024), which yields a model with increased diversity and blob adherence but decreased designability.

**Details of** PosEmbed**.** Our Invariant-Cross-Attention module in Algorithm 1 uses a positional encoding PosEmbed of the relative positions of ellipsoid means to residue positions. This is a sinusoidal positional encoding of the relative positions (the vectors between ellipsoid means and residue positions). Each number of the 3D offset vector is encoded into 64 dimensions, and all 3 are concatenated.

**Inference time protein length choice.** At inference time, ProtComposer (and Multiflow, RFDiffusion, and Chroma) take a protein length $L$ as input, which specifies the number of residues in the generated protein. In ProtComposer, each ellipsoid $\mathbf{E}_k$ is associated with a number of residues $n_k$ that is supposed to end up in that ellipsoid. The sum of $n_k$ does not have to match the total length $L$. The $n_k$ are just an additional conditioning input that the model can adhere to but does not have to. Even if the sum of $n_k$ is larger than $L$, the model can still (and empirically does) use some of the residues for strands and to connect the ellipsoids. When generating from ellipsoids from our ellipsoid statistical model, we set $L$ to be equal to the sum of $n_k$. When generating from data-extracted ellipsoids, we set $L$ to be the length of the original protein which the ellipsoids were extracted from, so $L > \sum n_k$.

## A.2 RUNNING CHROMA

For running Chroma (Ingraham et al., 2023), we use the author's GitHub. For Table 1, we run Chroma conditioned on the same data ellipsoids as ProtComposer, while the samples in the Pareto front (Figure 4) under varying sampling temperature are sampled unconditionally. The inverse-sampling temperatures that we sweep over are $[1, 1.4, 2, 4, 8, 10, 15, 20, 40, 80]$ where 10 is the default provided by the authors. The length distributions are the same as for Multiflow as described in A.4.

To condition Chroma, we utilize the conditioner classes provided in the author's GitHub repository. Concretely, we use the `ShapeConditioner` that is provided to condition on point clouds. The point cloud that we condition on is obtained by first sampling from a Gaussian mixture model that is defined by the set of means and covariances for the ellipsoid layout we condition on. Then, we reject or accept each sample based on whether or not its Mahalanobis distance to any of the ellipsoids' Gaussian is less than $\sqrt{5}\text{Å}$, which is the same threshold that we use to define our ellipsoid boundaries in equation 1, and that is used in our ellipsoid visualizations throughout the paper. The only parameter that we alter from the `ShapeConditioner` default settings is to set the number of residues according to our length distribution and to set `autoscale=False` since the desired protein volume is known and specified by the input point cloud.

## A.3 RUNNING RFDIFFUSION

For running RFdiffusion (Watson et al., 2023), we use the author's GitHub. The samples in the Pareto front (Figure 4) under varying sampling temperature are sampled unconditionally. The sampling temperatures that we sweep over are $[0, 0.2, 0.4, 0.6, 0.8, 1, 1.2, 1.4, 1.8, 2, 2.4, 2.8, 3.2, 4, 4.8, 6]$ where 1 is the default provided by the authors.

## A.4 PARETO FRONTIERS DETAILS

The black dot in Figure 4, which corresponds to the metrics achieved by PDB proteins from the PDB is computed from our validation set. This is the validation set of proteins that were deposited in PDB after 2021 used by Campbell et al. (2024).

**Running Multiflow and rotational annealing.** What we term "rotational annealing" is described as an exponential rate schedule for the rotation vector field in Multiflow (Campbell et al., 2024) and as "inference annealing" in FoldFlow (Bose et al., 2024). What we call rotational annealing strenghts is denoted $c$ in both of these papers. We run Multiflow at the rotational annealing strenghts $[0.0, 0.1, 0.2, 0.3, 0.4, 0.5, 0.6, 0.7, 0.8, 0.9, 1.0, 1.1, 1.2, 1.4, 1.6, 2, 3, 4, 5, 6, 10]$ since 4,5, and 6 all yield essentially identical results to the annealing strength 10 used in Multiflow by default, and variation is achieved at low annealing strenghts.

**Standard deviations in shaded regions.** The standard deviations that we show as shaded regions in Figure 4 are computed from 3 parameter settings for ProtComposer's frontiers and 3 for the Multiflow frontiers. We select the parameter settings for the point on the frontier that is at a designability of 0.98 for each of the three metrics. For all three parameter settings, we rerun inference with 4 additional seeds and compute the standard deviations from the results for all four metrics (designability, novelty, diversity, and helicity). The final reported standard deviations are obtained by averaging the standard deviations across the three points.

**Considering length distribution effects.** We chose a fixed number of ellipsoids $K$ for consistency across protein lengths and the specific value of $K = 5$ since it is the most frequent number of ellipsoids in PDB proteins (see Figure 11). However, this still gives rise to a distribution of lengths, and we detail our considerations to provide meaningful comparisons here: To construct ProtComposer's Pareto frontiers, we sweep over all parameter combinations of $\sigma, \nu, \gamma$ and $\lambda$. The parameters $\sigma, \nu, \gamma$ are parameters of ProtComposer's ellipsoid statistical model, with each combination giving rise to a different distribution of ellipsoids. Each ellipsoid distribution has a different length distribution (see Figure 8). Since designability is impacted by protein length and our evaluation metrics of helicity, diversity, and novelty could be impacted by length as well, we also sample Multiflow at all rotational annealing strengths with several length distributions. We combine all resulting points into a single

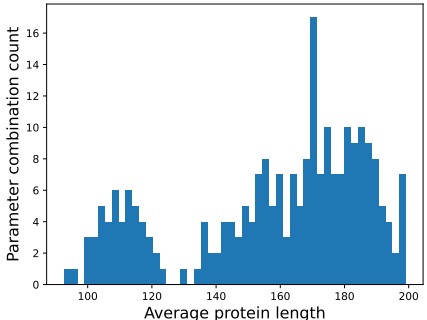 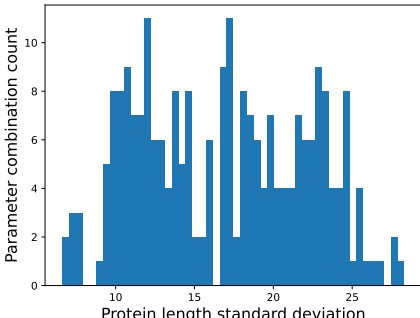

Figure 8: *Left:* Average lengths of proteins from our ellipsoid statistical model for all combinations of $\sigma \in [3, 4 \ldots 10]$, helix fraction $\gamma \in [0.2, 0.4, 0.6, 0.8, 1]$ and ellipsoid anisotropy $\nu \in [5, 10, 20, 50, 100]$. *Right:* The corresponding standard deviations of the length distributions of each parameter combination.

set and compute Multiflow's Pareto frontier from that set. In an attempt to cover the space of length distributions well, we sample with the following length distributions: 1) the length distribution of the $\sigma, \nu, \gamma$ combination with the minimum mean length. 2) the length distribution of the $\sigma, \nu, \gamma$ combination with the maximum mean lenght. 3) the length distribution of the $\sigma, \nu, \gamma$ combination with the average mean length. 4) a uniform length distribution between the minimum mean length of all length distributions (92) and the maximum (198).

**Designability computation.** We compute the fraction of structures for which at least one out of 8 sequences sampled by ProteinMPNN (Dauparas et al., 2022) results in scRMSD $< 2\text{Å}$ when refolded with ESMFold (Lin et al., 2023). We use the author's repositories for running the tools. For ProteinMPNN we use the backbone version that takes N, CA, C, O.

## B  DISCUSSION

**Additional Related Work.** Several other protein structure generative models were developed alongside, on top of, or after the mentioned Chroma (Ingraham et al., 2023), FrameDiff (Yim et al., 2023b), and RFDiffusion (Watson et al., 2023). This includes Genie (Lin & AlQuraishi, 2023), Anand & Achim (2022), Protpardelle (Chu et al., 2023), and Genie2 (Lin et al., 2024) among others. We also note DiffTopo (Miao & Correia, 2024), which first generates a skeleton protein representation, related to our ellipsoid layouts, where helices and individual strands of sheets are represented as three 3D points, then initializes existing secondary structure elements adhering to the skeleton, noises them, and then denoises them using RFDiffusion.

**User guidelines for choosing $K$:** All specifications of numbers of ellipsoids $K < 10$ can be expected to yield successful generations. To see which $K$ are the most in-distribution, please see the histogram in Figure 11 where we visualize the frequency of different $K$.

**Design of ICA and treating ellipsoids equivalent to residue frames.** In our transformer layers (algorithm 2), the residue tokens update all ellipsoid tokens, and ellipsoid tokens update all residue tokens. In Invariant-Cross-Attention (algorithm 1), we inject ellipsoid position and geometry information into the residues tokens without updating ellipsoid tokens.

The reason: ICA updates a residue token based on transforming the ellipsoid means and covarianc matrices into the local coordinate frame of the residue (where residue frames are defined as in AlphaFold2). The same mechanism is not applicable to updating ellipsoid tokens based on residue positions since a canonical assignment of a frame to an ellipsoid is not possible. To see this, we consider the worst-case scenario of a round ellipsoid - clearly, no canonical assignment of a 3D orientation is possible. In the best-case scenario of an ellipsoid with three distinctly sized principal components, we could choose, e.g., the two largest to construct a frame from. However, their sign is arbitrary, leading to 4 options among which no canonical choice exists. Thus, we opted for our ICA without ellipsoid token updates, which is empirically sufficient for strong ellipsoid adherence.

Furthermore, only injecting relative positional encoding into the keys and values and not the queries was our default choice since it is how relative positional encodings are used in language model transformers (Shaw et al., 2018) or in geometric transformers such as SE3-Transformer (Fuchs et al., 2020).

**Specifying the number of residues per ellipsoid.** In our Ellipsoid definition, each ellipsoid $k$ is annotated with a number of residues $n_k$ that should be associated with the ellipsoid. Removing this additional feature could provide additional flexibility for the model at inference time. We chose to specify the number of residues per ellipsoid to give the user the option of controlling it (or having the number be determined via linear fit if preferred).

**Possible ProtComposer use cases.** We envision frameworks based on ProtComposer where only secondary structure conditioning is possible to enable use cases such as the following:

- Example use case: We aim to scaffold a therapeutically relevant functional site. The protein requires a certain shape to fit into a delivery mechanism. With ProtComposer we can specify the rough shape and size of the scaffold to still fit into the delivery mechanism

- ProtComposer can redesign the connectivity of secondary structure elements: biologists aim to escape the existing space of protein topologies and discover new ones that can be used as scaffolds or for other design tasks.

- Example use case: We aim to design a binder for a target at a flat beta-sheet region. With ProtComposer, we can specify that a beta-sheet of the right size and shape should interface with the target's beta-sheet to increase the probability of success in generating a strong binder.

- We often know how much flexibility/rigidity we want in certain areas of the protein. With ProtComposer, we can place a rigid helix bundle, a beta-barrel, or more loosely connected substructures in those regions.

**The Compositionality Metric.** We note that our compositionality metric, the *effective number of components*, which is $\exp(-\sum_k p_k \log p_k)$ where $p_k = m_k / \sum_k m_k$, is based on the *Diversity Index* which is commonly employed in ecology or demography.

## C  ADDITIONAL RESULTS

### C.1  MULTIFLOW CODESIGN SEQUENCES

Table 3: **Sequence-structure co-generation.** ProtComposer's self-consistency when generating sequence and structure jointly vs. dropping its generated sequence and producing the sequence with ProteinMPNN. 1-seq Designability refers to generating 1 sequence per structure, while 8-seq Designability uses the best out of 8 sequences per structure. Self-consistency RMSD is abbreviated as scRMSD.

| Approach | 1-seq Designability↑ | Median scRMSD↓ | Mean scRMSD↓ | 8-seq Designability↑ |
|---|---|---|---|---|
| Joint Generation | 0.75 | 1.76 | 2.15 | – |
| ProteinMPNN | 0.81 | 1.65 | 2.41 | 0.98 |

**Multiflow codesign sequences.** Whenever assessing designability we use ProteinMPNN to infer the sequence for a protein structure generative model's generated sequence. Our base model Multiflow jointly generates a protein sequence and structure. Commonly we ignore this generated sequence and also produce a sequence conditioned on the structure using ProteinMPNN. Here we assess how ProtComposer's co-generated sequences compare with those obtained by ProteinMPNN conditioned on ProtComposer's structures.

For this purpose, we select statistical model parameters with high designability ($\nu = 50, \sigma = 5$) and draw 400 structures together with sequences from ProtComposer. Table 3 shows that the designability of the jointly generated sequence is lower than that of a single sequence generated with ProteinMPNN (the default designability metric shows the best of 8 ProteinMPNN sequences). This is similar to the Multiflow paper (Campbell et al., 2024), where co-design did not provide designability

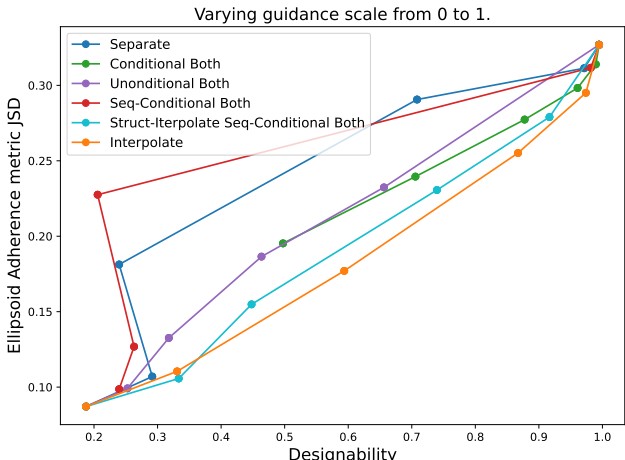

Figure 9: Evaluation of our different self-conditioning variants in terms of designability and ellipsoid adherence measured in terms of Resegment JSD. The different variants are described in Appendix C.2.

improvements. Interestingly, the median self-consistency RMDSs (scRMSD) of the jointly generated sequences are worse, while their mean scRMSDs are better.

## C.2 ABLATION OF VARIANTS FOR SELF-CONDITIONING UNDER GUIDANCE

Recall that both Multiflow and ProtComposer use *self-conditioning* (Chen et al., 2023), in which, during inference, the flow-model receives the output of the previous integration step as additional *self-conditioning input*. During inference, the unconditional model $p_\theta(\mathbf{t}, R, \mathbf{a})$ produces the self-conditioning variable $X$, and from the ellipsoid conditioned model $p_\theta(\mathbf{t}, R, \mathbf{a} \mid \mathbf{E})$, we obtain $X_\mathbf{E}$. Instead of supplying $X$ to the unconditional and $X_\mathbf{E}$ to the conditioned model, we use $\lambda X_\mathbf{E} + (1 - \lambda)X$. Please refer to the Multiflow paper (Campbell et al., 2024) for information on how Multiflow without classifier-free guidance performs self-conditioning.

This choice is based on our exploration of self-conditioning variants that are shown in Figure 9, which shows that the *interpolate* option performs best. The variants we explore are:

- *Separate:* supply $X$ to the unconditional and $X_\mathbf{E}$ to the conditioned model.
- *Conditional Both:* supply $X$ to both models.
- *Unconditional Both:* supply $X_\mathbf{E}$ to both models.
- *Seq-Conditional Both:* supply $X$ to the unconditional and $X_\mathbf{E}$ to the conditioned model for the structure self-conditioning input while the sequence self-conditioning input is extracted from $X_\mathbf{E}$ for both models.
- *Struct-Iterpolate Seq-Conditional Both:* supply $\lambda X_\mathbf{E} + (1 - \lambda)X$ to both models for the structure self-conditioning input while the sequence self-conditioning input is extracted from $X_\mathbf{E}$ for both models.
- *Interpolate:* supply $\lambda X_\mathbf{E} + (1 - \lambda)X$ to both models.

## C.3 FURTHER RESULTS

Table 4: Compositionality of different protein structure generative models.

|  | ProtComposer ($\lambda = 2.0$) | Multiflow | Chroma | RFDiffusion |
|---|---|---|---|---|
| Compositionality | 3.2 | 1.9 | 2.5 | 3.3 |

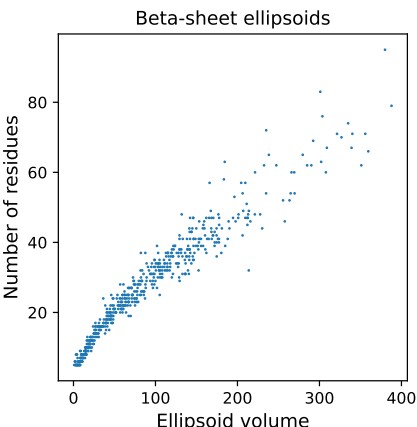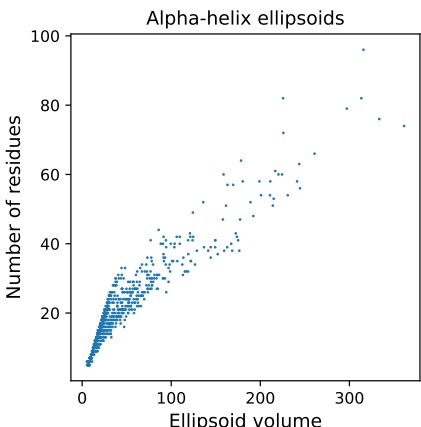

Figure 10: Scatter plots of the number of residues in an ellipsoid and the ellipsoid's volume for beta-sheet and helix ellipsoids. The relationship is close to linear. Thus, we choose the number of residues for an ellipsoid from our synthetic ellipsoid statistical model based on their volume and the linear fit to residue count. For beta sheet ellipsoids, the linear fit correlation coefficient is 0.93, and the p-value is 0.0. For alpha-helix ellipsoids, the linear fit is 0.97, and the p-value is 0.0.

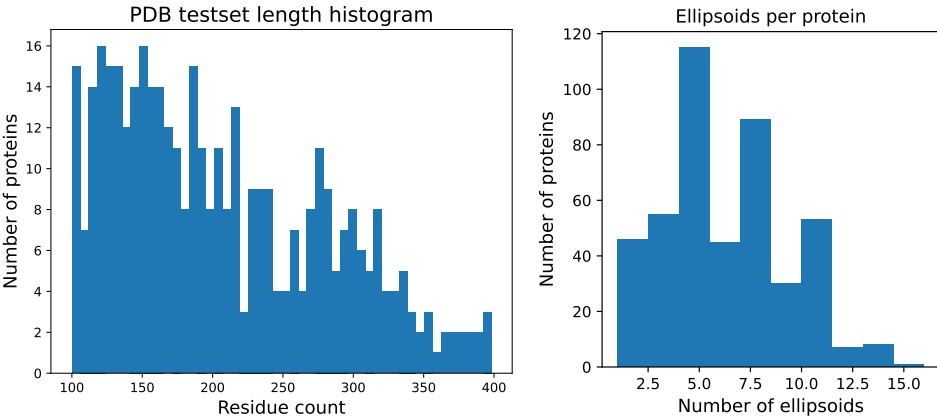

Figure 11: Histograms for the validation set of PDB proteins. *Left:* Histogram of protein lengths. *Right:* Histogram of the number of ellipsoids per protein with our default segmentation parameters.

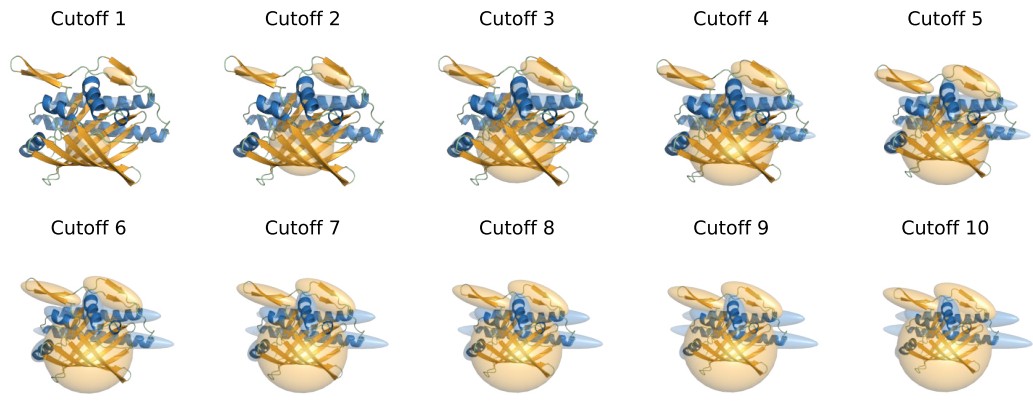

Figure 12: Different cutoffs of squared Mahalanobis distance for visualizing ellipsoids.

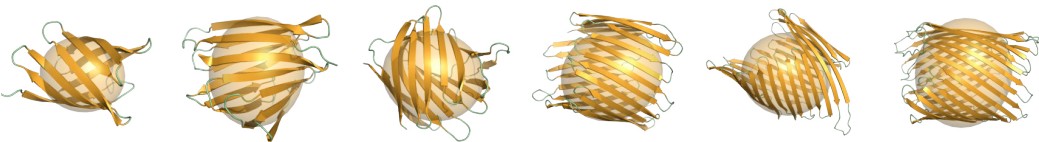

Figure 13: Hand-specified beta-barrel of increasing size.

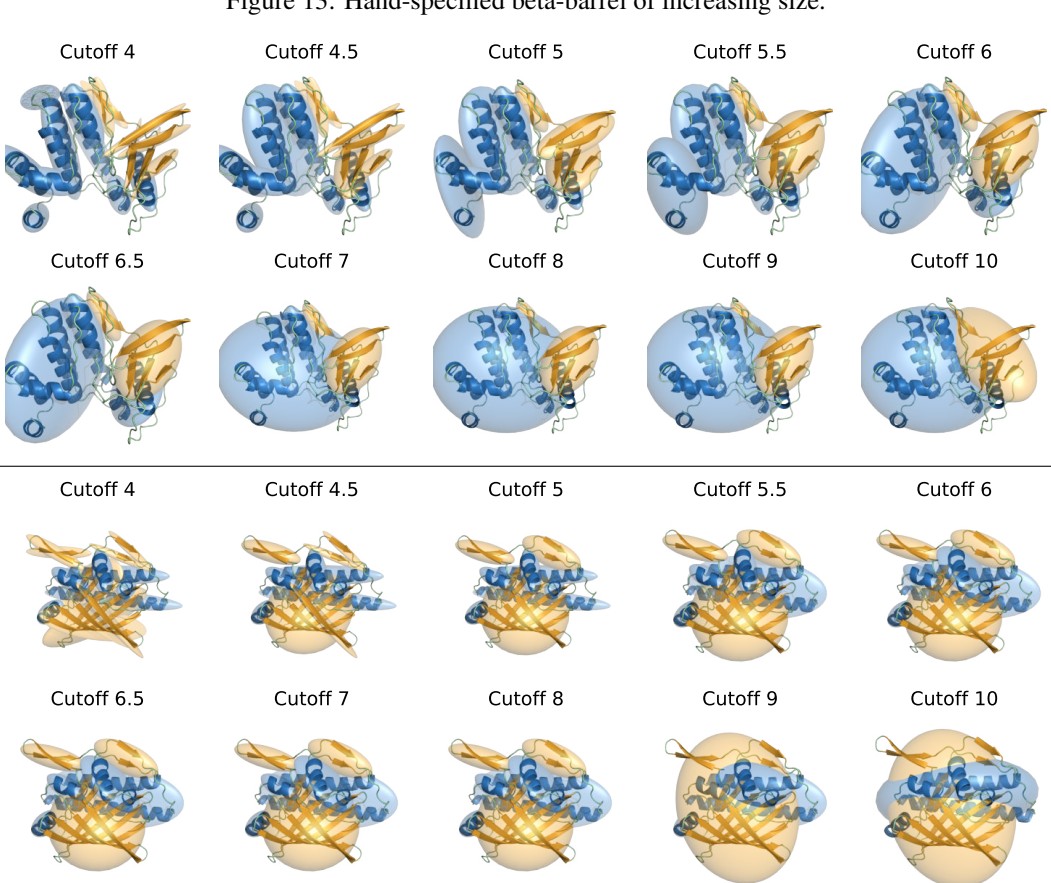

Figure 14: Ellipsoids obtained with different radius cutoffs (Å units) for the residue segmentation (Algorithm 3) for the two proteins with PDB IDs 7DG4 (top) and 7V2T (bottom).

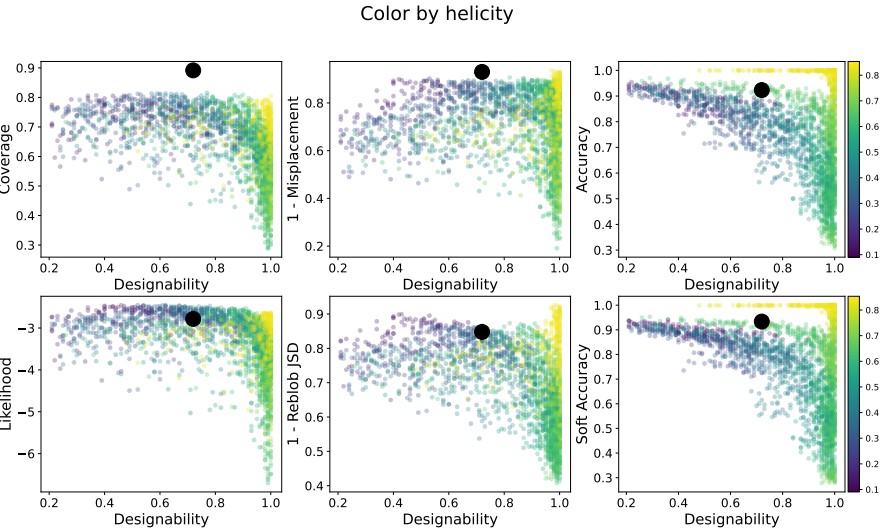

Figure 15: Scatter plots of designability vs. ellipsoid adherence metrics when conditioning on synthetic ellipsoids that were drawn from our ellipsoid statistical model.

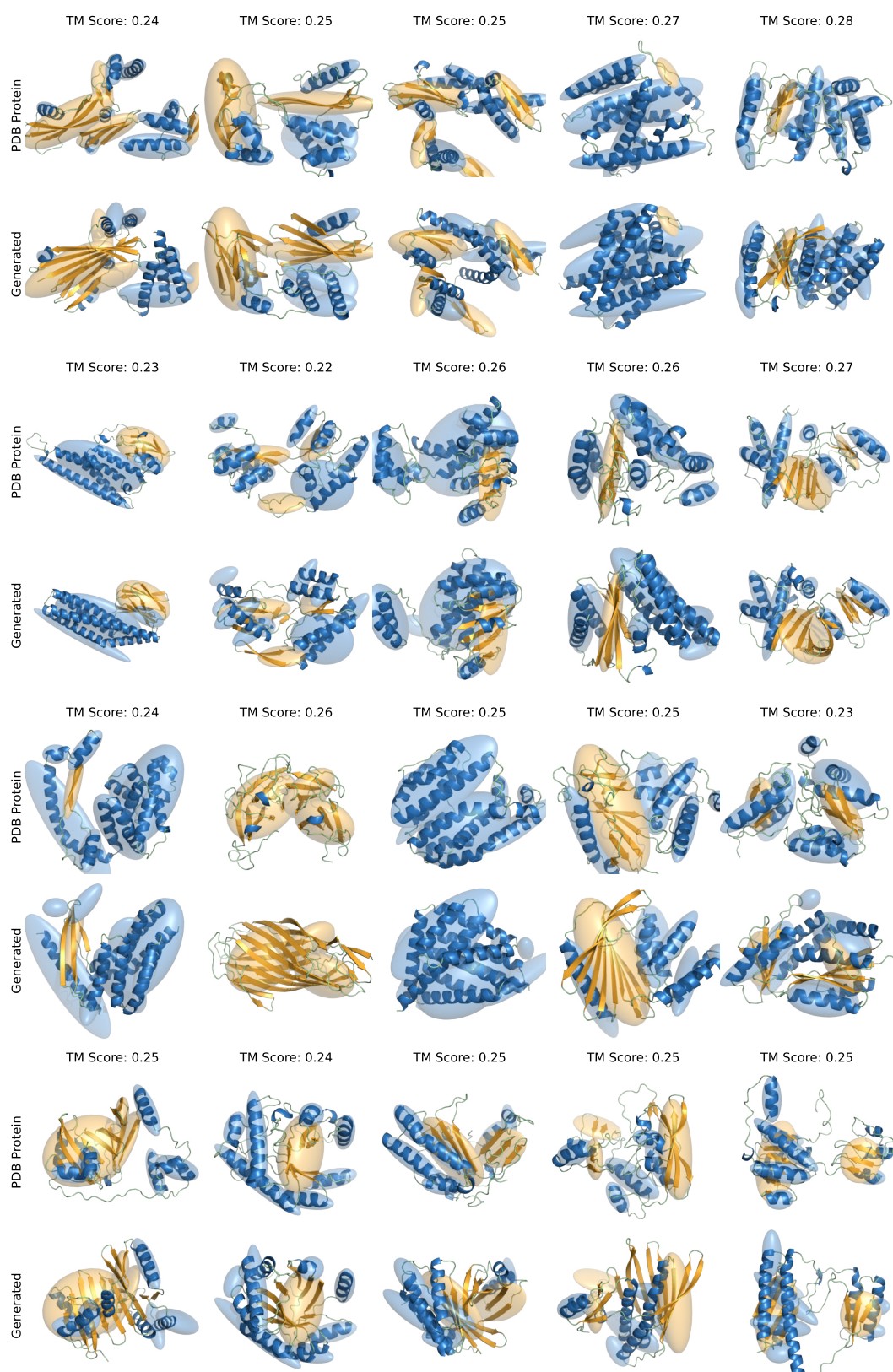

Figure 16: Several rows of randomly chosen protein structures from the PDB validation set, their corresponding ellipsoids, and, below them, the protein we generate conditioned on those ellipsoids. Each pair of PDB protein and generated protein is annotated with the TM-Score between them. **The TM-Scores are very low and we generate novel proteins while adhering to the layouts.**

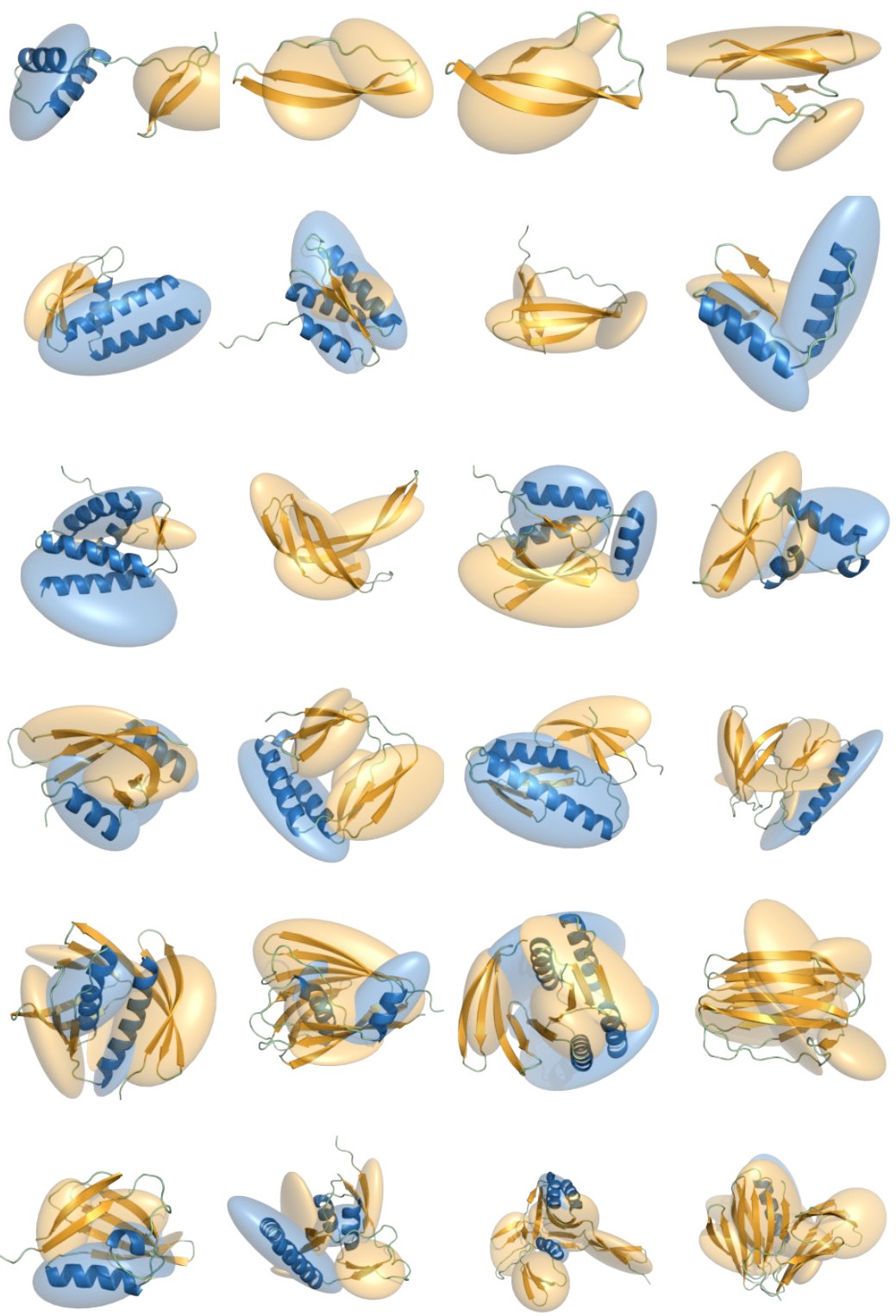

Figure 17: Random samples of ellipsoids and the generated proteins from our ellipsoid statistical model with parameters $\sigma = 6$, $\nu = 5$, $\gamma = 0.4$, and the number of ellipsoids $K$ varying from 2 (top) to 7 (bottom).

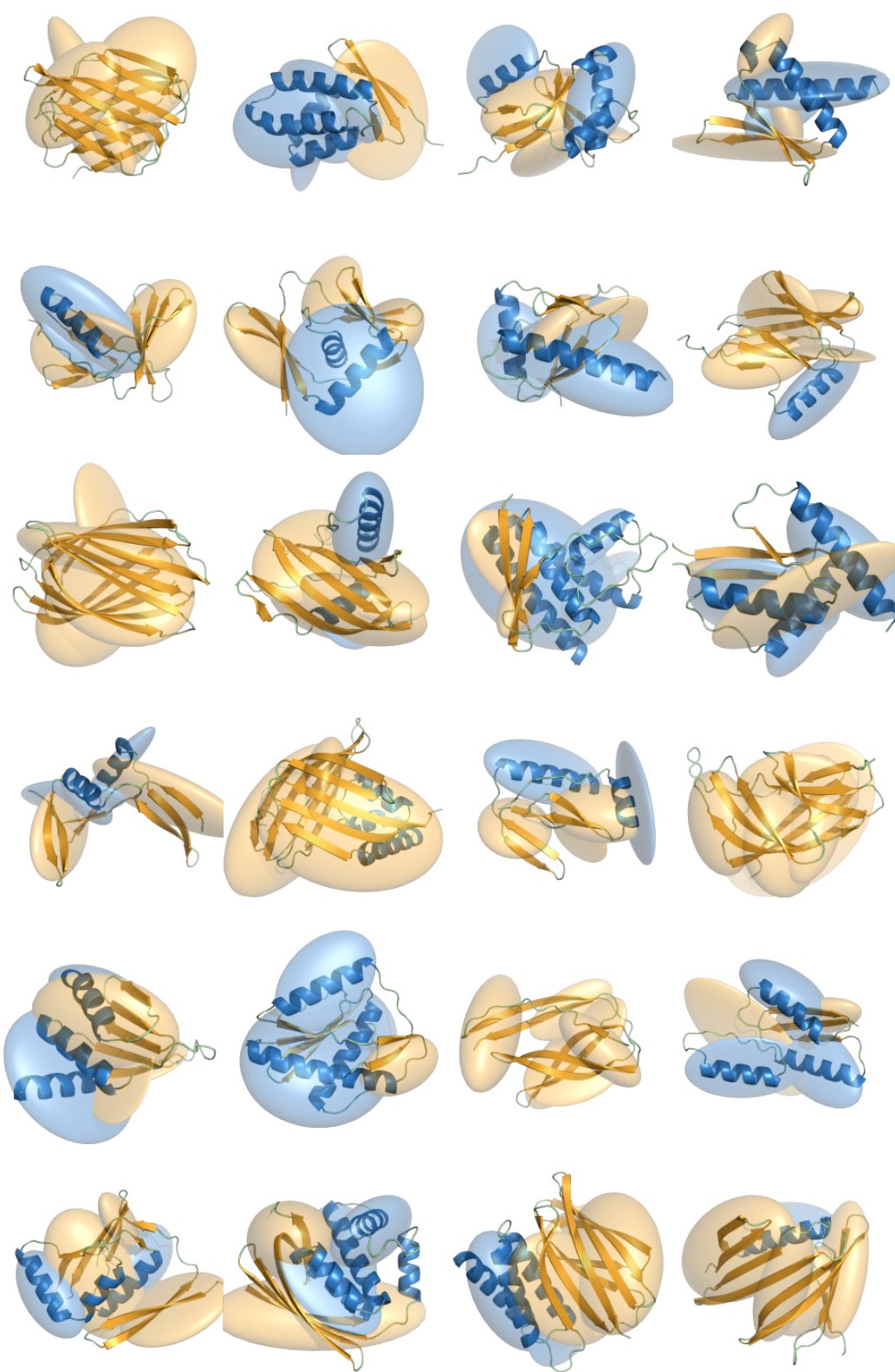

Figure 18: Random samples of ellipsoids and the generated proteins from our ellipsoid statistical model with parameters $\sigma = 6, \nu = 5, \gamma = 0.4$, and the number of ellipsoids being $K = 5$.

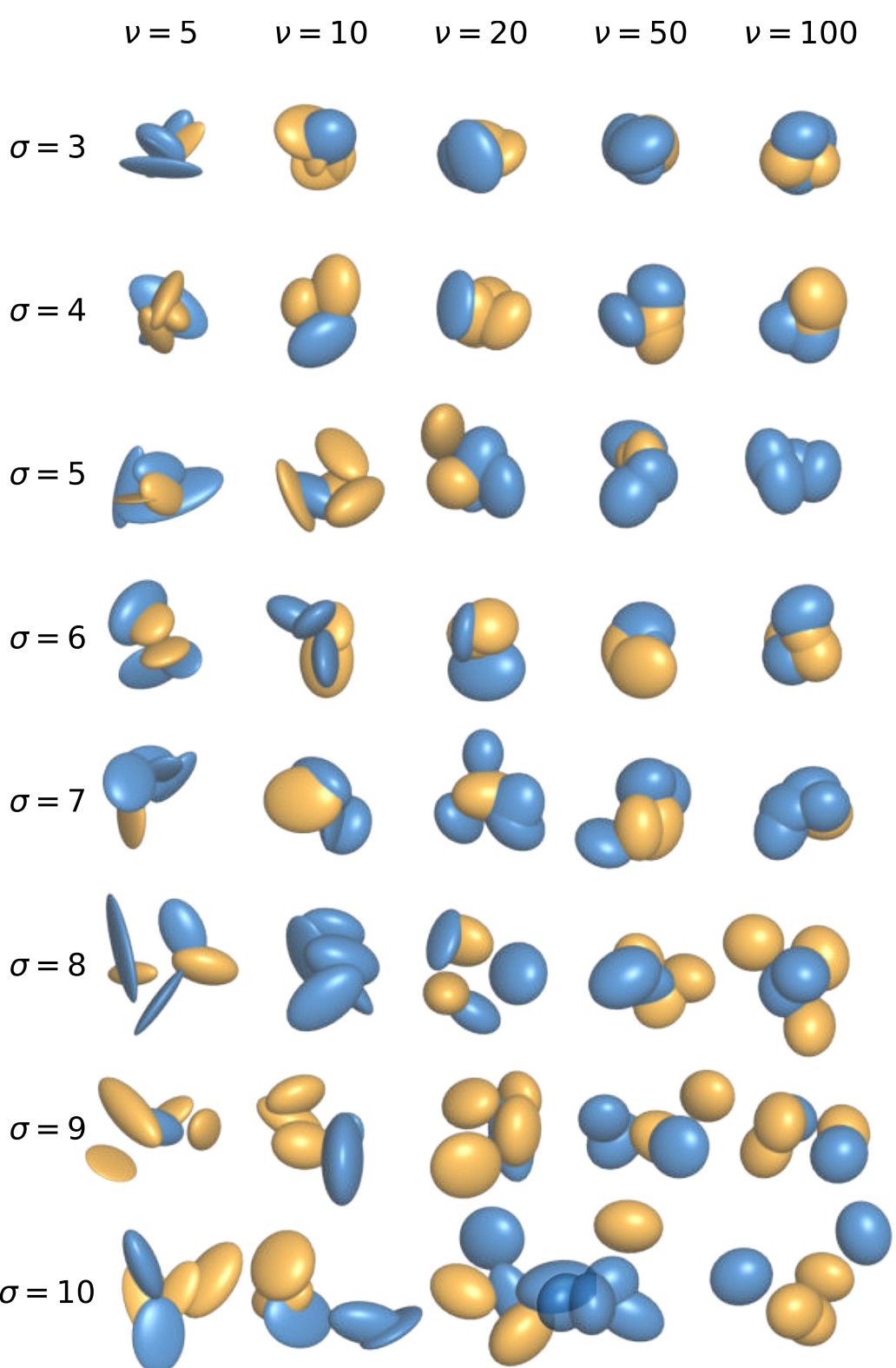

Figure 19: Ellipsoids sampled from our ellipsoid statistical model with different parameter combinations of $\nu$ and $\sigma$, which control the ellipsoids' "roundness" or anisotropy and their compactness, respectively.

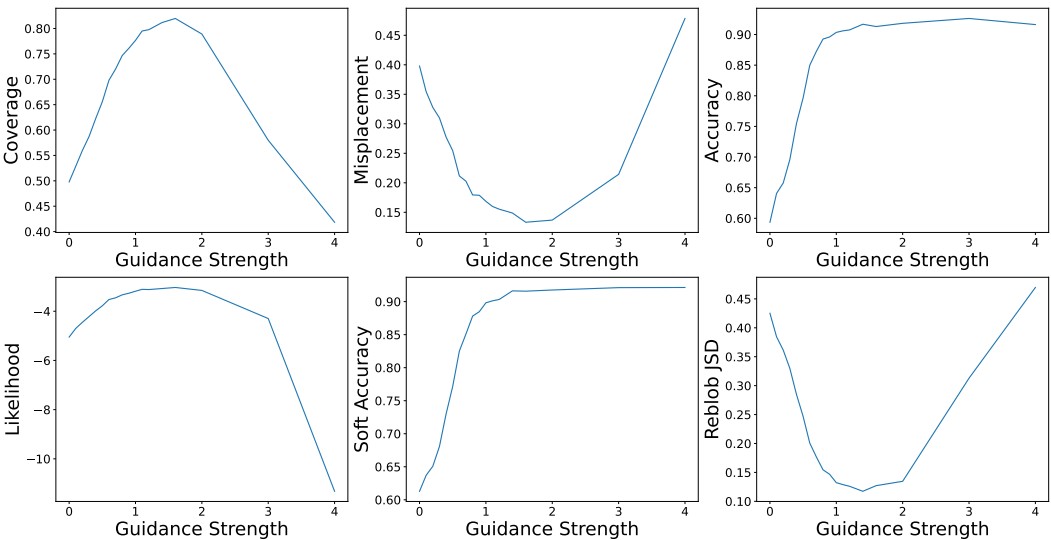

Figure 20: Ellipsoid adherence metrics under varying scales of guidance strength $\lambda$ for ellipsoids extracted from the validation data. The highest ellipsoid adherence is attained at values of $\lambda > 1$.

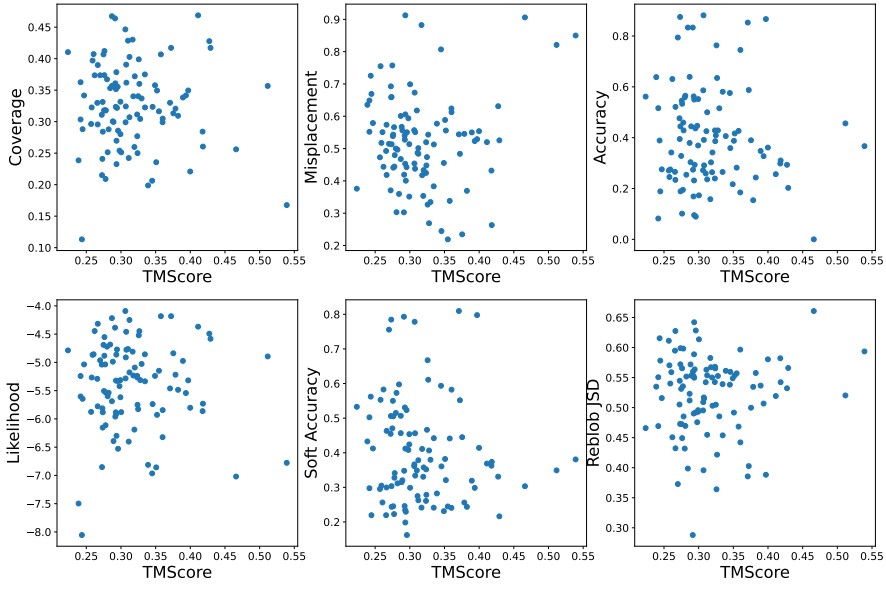

Figure 21: We generate proteins conditioned on ellipsoids extracted from the validation data and show scatter plots of ellipsoid alignment metrics and the TMScore between the PDB protein and the generated protein. In general, the TMScore is very low, and the generated proteins are novel.

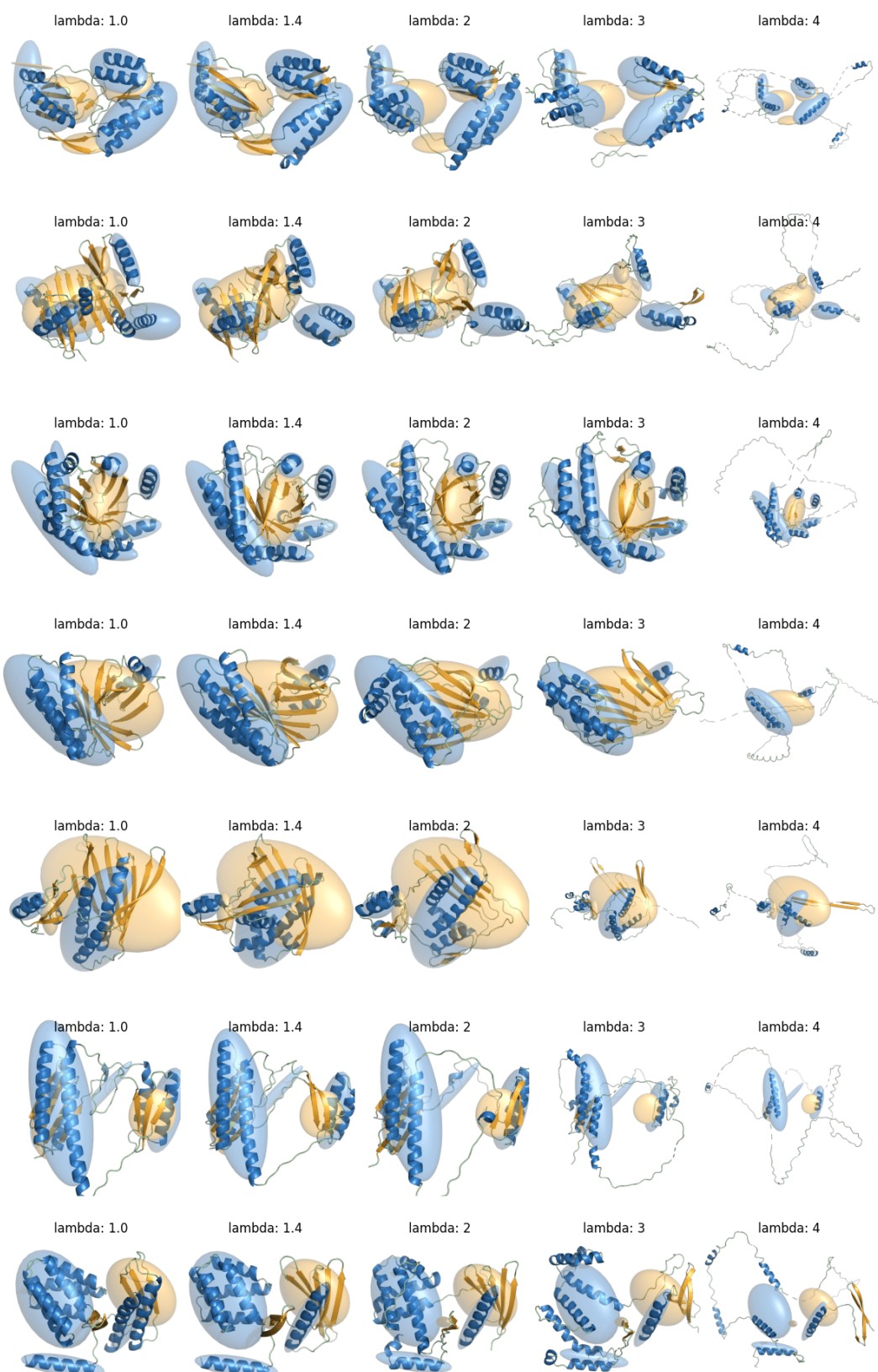

Figure 22: Several protein layouts (rows) and ProtComposer generations for them with varying guidance (columns) where guidance strengths ($\lambda \geq 1$) including extreme values for which the model breaks down.

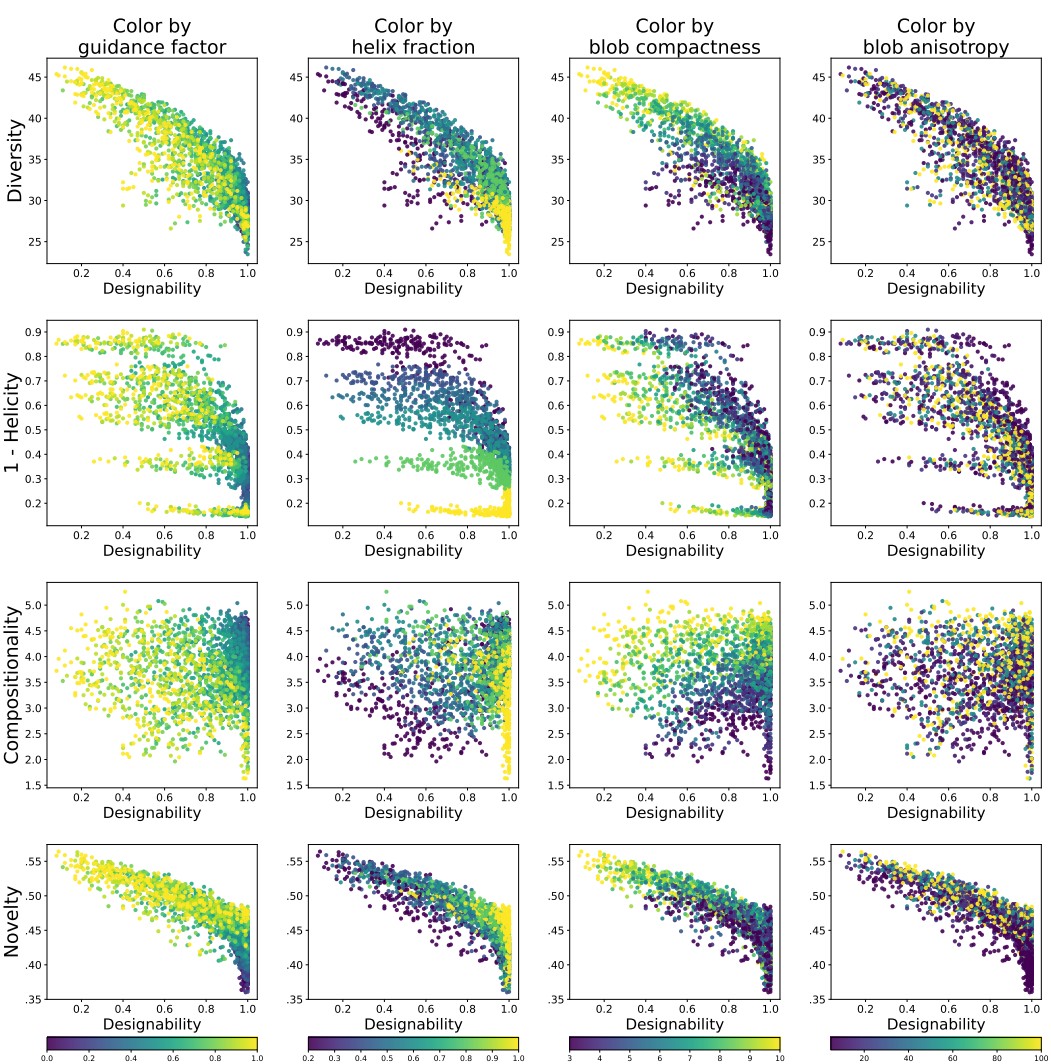

Figure 23: Scatter plots of designability and novelty, entropy, helicity, or diversity when conditioning on synthetic ellipsoids that were drawn from our ellipsoid statistical model. The colors indicate the value of the statistical model's parameter that is in the caption of each column.

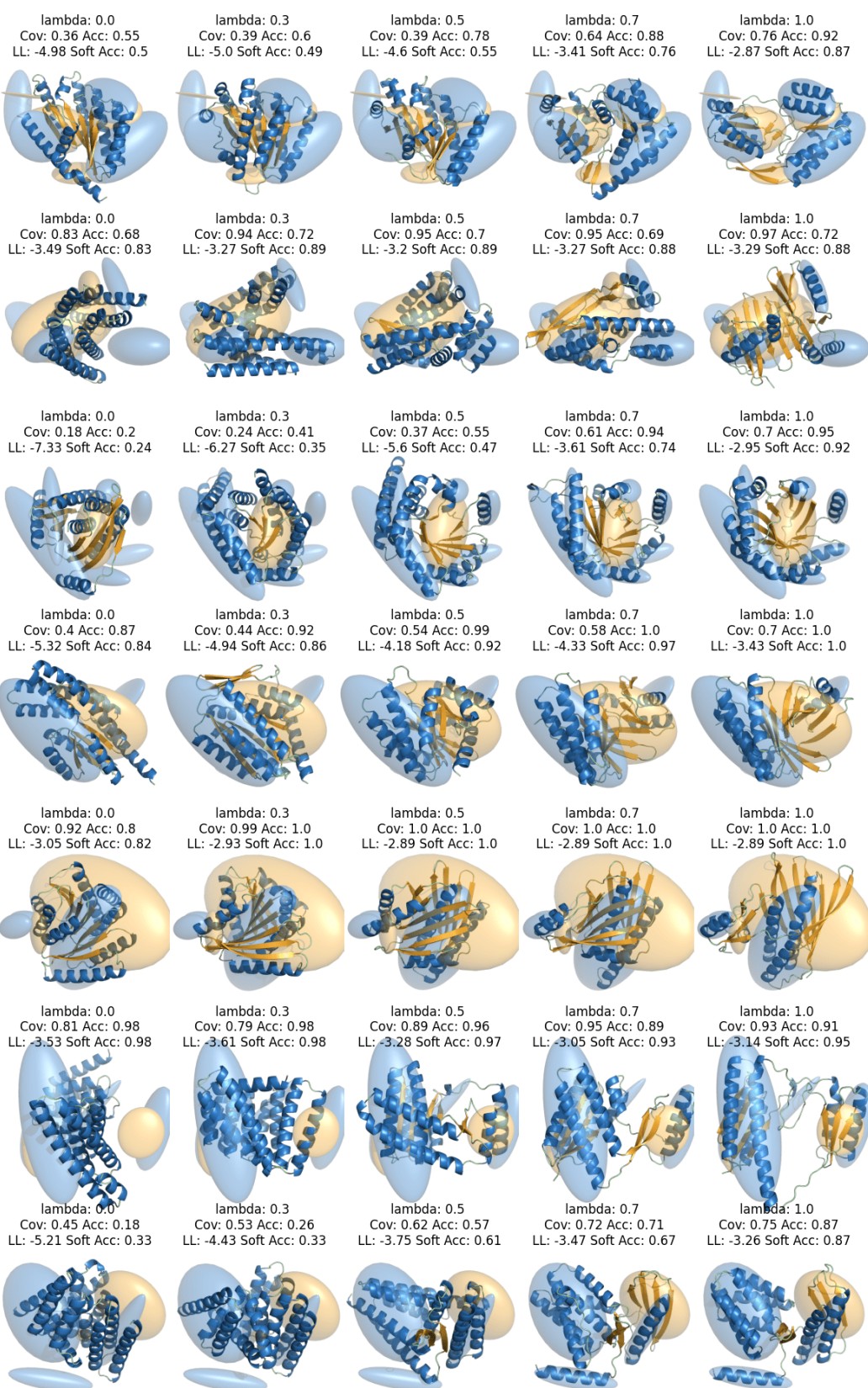

Figure 24: Several protein layouts (rows) and ProtComposer generations for them with varying guidance (columns); no guidance ($\lambda = 0$) on the left, full guidance ($\lambda = 1$) on the right.

