# OpenReview forum: "ProtComposer: Compositional Protein Structure Generation with 3D Ellipsoids"
_ICLR.cc/2025/Conference — ICLR 2025 Oral_

### Official Review · Reviewer_eiq5 · 2024-11-02

**Soundness:** 3
**Presentation:** 3
**Contribution:** 3
**Rating:** 8
**Confidence:** 4

**Summary:**

Introduces ProtComposer, a generative model for proteins. ProtComposer seeks better control over the shape of generated proteins, as well as allow for greater novelty in the generation of proteins. Notably, the user has to choose between control or novelty, ProtComposer does not achieve both simultaneously. These tasks are accomplished by the introduction of 3D ellipsoid frames to guide the generation of proteins by the pre-existing Multiflow algorithm.

Without a pre-existing metric the authors feel sufficiently quantifies compositionality, they introduce their own metric. It is shown that ProtComposer outperforms existing methods in this metric.

**Strengths:**

It addresses an important and relevant area. The results are overall quite impressive as well. Additionally, showing the ability to work with handcrafted ellipsoid frames or the more easily scalable ML generated frames shows the practicality of ProtComposer.

I really like the point about using simple ML models for ellipsoid sampling as compared to NNs. I would like to see some stronger analytical or experimental justification of the claim though.

**Weaknesses:**

Since a custom metric is introduced in this paper and then used as justification for the performance of the model, a section (either in the main paper or the appendix) justifying this metric would be nice. Showing comparisons to other pre-existing metrics, performance of many other algorithms under this metric, or stronger domain justification would strengthen the meaning of the results. I did not reject over this since the metric intuitively looks good, but further empirical or analytical support would be nice.

The formatting of the extra results in the appendices results in very odd page layouts, where some pages are blank, others have odd whitespace gaps, etc. Adjustment to make these pages more presentable would be nice.

**Questions:**

I am assuming that the user can redefine K during each trial as they see fit, though studies on guidelines in choosing K would be helpful. In situations where the user has only a vague idea of what they are looking for (an unfortunately common occurrence), having a guide on where to start would be beneficial.

---

> ### Author Response · Authors · 2024-11-17
> **Response by Authors**
>
> Thank you for the review! To address your questions and concerns (our updated manuscript can be downloaded at the top of this page):
>
> ---
>
> **Regarding introduced metric: "other algorithms under this metric, or stronger domain justification"**
>
> We assume that you refer to our compositionality metric and are glad you find it intuitive! We added Table 4 to the revised manuscript, where we evaluate additional protein structure generative models' compositionality. We note that an ellipsoid conditioned model could be built out of any protein structure generative model, and the base model does not necessarily have to be Multiflow.
>
> |                    | ProtComposer ($\lambda=2.0$) | Multiflow | Chroma | RFDiffusion |
> |--------------------|----------------------------|-----------|--------|-------------|
> | Compositionality   | 3.2                        | 1.9       | 2.5    | 3.3         |
>
> Further, we add an explanation to Appendix B: that our compositionality metric is inspired by the ``Diversity Index" (https://en.wikipedia.org/wiki/Diversity_index), which is often used in the domains of ecology or demography.
>
> **Formatting and white space in the appendix**
>
> Thank you for pointing it out! We fixed it in our revision.
>
> **User guidelines for choosing the number of ellipsoids $K$**
>
> That is a nice suggestion! We added Figure 11 with a histogram of $K$ for PDB proteins and this guidance to Section B: All specifications of numbers of ellipsoids $K<10$ can be expected to yield successful generations. To see which $K$ are the most in-distribution, please see the histogram in Figure 11 where we visualize the frequency of different $K$.
>
> ---
>
> We hope the discussions and results address your concerns! Please let us know if there are any further opportunities to improve the score.

---

> > ### Comment · Reviewer_eiq5 · 2024-11-18
> > **Consideration of Updates**
> >
> > Thank you for the excellent work and updates.
> >
> > My main concern was a justification of the metric's validity given that it is introduced in this paper. I am wary of new, custom metrics which are used to show how well the new technique/model performs. However, the comparison with the Diversity Index as well as (to a lesser degree) the added table alleviate my concerns about the metric's origins. I am satisfied with this.
> >
> > I had concerns over how to choose K in practice. The results provided in the appendix help give a user a good starting point for experimentation, so I am also satisfied with the response to this concern.
> >
> > Overall, I think that this is a strong submission. I am happy to give it a high score!

---

> > > ### Author Response · Authors · 2024-11-22
> > > **Response by Authors 2**
> > >
> > > It is great to hear that all your concerns are resolved - thank you for your help in improving the paper and for the thoughtful points!
> > >
> > > Since we are excited that you, as well as the other reviewers, consider our work excellent and as a strong submission, please let us know if there is anything we can do to further improve the paper and make you consider modifying and raising your paper rating from **“8: accept, good paper”** to **“10: strong accept, should be highlighted at the conference”**, or if you think such a score increase is already warranted.

---

> > > > ### Author Response · Authors · 2024-11-25
> > > > **Response by Authors 3**
> > > >
> > > > With the discussion period ending tomorrow, we thank you for the work together toward a better paper! We would be excited to hear any further feedback on our second response above if you can find the time!

---

> > > > > ### Comment · Reviewer_eiq5 · 2024-11-25
> > > > > **Consideration of Score**
> > > > >
> > > > > I appreciate the enthusiasm of the authors with regards to the revision process. I think the paper is very interesting and a strong addition to Comp Bio literature. I reserve a score of 10 for papers I believe will become instrumental throughout the field and lead to the creation of a new standard set of practices in the field. I think further fine-grained control over the generation process would be needed for me to change the score from an 8 to a 10. Nonetheless, I think it is a very strong paper and represents excellent research.

---

### Official Review · Reviewer_D7Wa · 2024-11-03

**Soundness:** 4
**Presentation:** 4
**Contribution:** 4
**Rating:** 8
**Confidence:** 4

**Summary:**

This paper presents ProtComposer, which is a fine-tuned model from MultiFlow to support conditional generation based on 3D ellipsoids, showing success at controllable design and achieving SOTA on the Pareto frontier of designability and diversity.

**Strengths:**

The paper is well written with clean visualizations demonstrating methods and results. The concept of utilizing ellipsoids as conditions for protein generation is interesting and novel, providing a bridge between protein-level conditioning and atom-level conditioning. In addition, the authors propose an effective Invariant Cross Attention module for integrating ellipsoid conditioning and demonstrate success at achieving SOTA performance on the Pareto frontier of designability and diversity.

**Weaknesses:**

I have no major concerns about this paper. However, it would be helpful if the authors could elaborate on the applicability of the ellipsoid-based conditioning approach on practical protein design tasks. How would it help with or facilitate the protein design process?

**Questions:**

- Line 355: How is the length between ellipsoids determined/sampled? Also, consider an ellipsoid with beta strand annotation, how is the length between each stranded segment determined (particularly if a strand ellipsoid is formed by segments that are distant in sequence but close in structure)?
- It would be helpful if the authors could provide ablation study results on the effect of self-conditioning, particularly the two self-conditioning schemes described in line 290.
- Figure 9: What is the linear fit and statistical significance in both cases?
- During training, is the ellipsoid conditioning information always provided, or only provided for a percentage of time?
- Line 367: Why is a structured residue considered as “covered” if it is inside at least one ellipsoid instead of inside the ellipsoid it is assigned?
- Table 1: Could the authors provide some intuition on why over-guidance (\lambda > 1) performs better than the conditional model itself (\lambda = 1)?
- Table 2: Does “PDB proteins” correspond to the validation dataset or does it include the training dataset?

---

> ### Author Response · Authors · 2024-11-17
> **Response by Authors**
>
> Thank you for the review! To address your questions and concerns (our updated manuscript can be downloaded at the top of this page):
>
> ---
>
> **On the applicability of the ellipsoid conditioning on practical protein design tasks**
>
> An important question! We added the following to Appendix B:
> - Example use case: We aim to scaffold a therapeutically relevant functional site. The protein requires a certain shape to fit into a delivery mechanism. With ProtComposer we can specify the rough shape and size of the scaffold to still fit into the delivery mechanism
> - ProtComposer can redesign the connectivity of secondary structure elements: biologists aim to escape the existing space of protein topologies and discover new ones that can be used as scaffolds or for other design tasks.
> - Example use case: We aim to design a binder for a target at a flat beta-sheet region. With ProtComposer, we can specify that a beta-sheet of the right size and shape should interface with the target's beta-sheet to increase the probability of success in generating a strong binder.
> - We often know how much flexibility/rigidity we want in certain areas of the protein. With ProtComposer, we can place a rigid helix bundle, a beta-barrel, or more loosely connected substructures in those regions.
>
>
>
> **How is the length between ellipsoids determined/sampled?**
>
> We add the following clarification to A.1:
>
> At inference time, ProtComposer (and Multiflow, RFDiffusion, and Chroma) take a protein length $L$ as input which specifies the number of residues in the generated protein. In ProtComposer, each ellipsoid $\mathbf{E}_k$ is associated with a number of residues $n_k$ that is supposed to end up in that ellipsoid. The sum of $n_k$ does not have to match the total lenght $L$. The $n_k$ are just an additional conditioning input which the model can adhere to but does not have to. Even if the sum of $n_k$ is larger than $L$, the model can still (and does) use some of the residues for strands and to connect the ellipsoids. When generating from synthetic ellipsoids, we set $L$ to be equal to the sum of $n_k$. When generating from data-extracted ellipsoids, we set $L$ to be the length of the original protein which the ellipsoids were extracted from, so $L$ > $\sum n_k$.
>
> **"provide ablation study results on the effect of self-conditioning, particularly the two self-conditioning schemes described in line 290."**
>
> We added Section C.2 with Figure 9 to the revised manuscript which shows the designability vs. ellipsoid adherence frontier under our 5 different possibilities of performing self-conditioning, showing that the interpolation variant performs best under all settings of guidance strength.
>
> **Figure 9: What is the linear fit and statistical significance in both cases?**
>
> We added to the Figure's caption that the linear fit for alpha-helices is 0.97 and for beta-sheets 0.93 and that for both, the statistical significance is high with a p-value that is so close to 0 as to be outside of numerical precision.
>
> **"During training, is the ellipsoid conditioning information always provided, or only provided for a percentage of time?"**
>
> For classifier free guidance we combine a conditional and an unconditional model. During pretraining, to obtain the unconditional Multiflow, there is never any ellipsoid conditioning. When fine-tuning Multiflow to obtain the conditional model, ellipsoid conditioning is always provided.
>
> **Why is a structured residue considered as “covered” if it is inside at least one ellipsoid instead of inside the ellipsoid it is assigned?**
>
> The notion of a residue being assigned to an ellipsoid does not exist at inference time. The ellipsoids only have an assigned number of residues, no assignment of which residue will go into them. The model can freely choose which residue ends up in which ellipsoid.
>
> **Intuition on why "over-guidance" with $\lambda > 1$ can yield improvements (Table 1)**
>
> A very interesting question! This is a long-standing and not fully understood phenomenon of diffusion/flow models. Figure 2 in the classifier-free guidance paper (https://arxiv.org/pdf/2207.12598) provides intuition by showing what happens for a Gaussian mixture - with "over-guidance" we make "extra-sure" that the samples are close to the conditioning distribution, and far from other possibilities. Most justification is empirical and, e.g., this blog post's Section 3 (https://sander.ai/2022/05/26/guidance.html) shows how image quality is improved with "over-guidance" where $\lambda = 3$. Lastly, the autoguidance paper (https://arxiv.org/pdf/2406.02507v1) observes that improvements are even possible when guiding a model with a worse, less trained version of itself.
>
> **Whether our "PDB protein" evaluation set is included in the training data**
>
> This set of proteins is separate from the training data.
>
> ---
> We hope the discussions and results address your concerns! Please let us know if there are any further opportunities to improve the score.

---

> > ### Author Response · Authors · 2024-11-25
> > **Response by authors 2**
> >
> > With the discussion period ending tomorrow, we thank you for the work together toward a better paper!
> >
> > Since we are excited that you, as well as all other reviewers, consider our work as a good submission, please let us know if there is anything we can do to further improve the paper and make you consider modifying and raising your paper rating from “8: accept, good paper” to “10: strong accept, should be highlighted at the conference”, or if you think such a score increase is already warranted.

---

> > > ### Comment · Reviewer_D7Wa · 2024-12-02
> > > **Thank you for the response**
> > >
> > > Thank you for your response and all my questions are addressed. I will keep my score since I reserve 10 for fundamental work that can have significant impacts across multiple fields.

---

### Official Review · Reviewer_679L · 2024-11-03

**Soundness:** 3
**Presentation:** 4
**Contribution:** 3
**Rating:** 8
**Confidence:** 3

**Summary:**

This work extends Multiflow to accept spatial conditioning of secondary structures via 3D ellipsoids, aiming to improve control in protein generation and reduce the overrepresentation of alpha-helices in current generative models. Building on Multiflow’s architecture, the authors address two main challenges:

1. Integrating and updating ellipsoid conditioning with structure embeddings with minimal modifications: they introduced an *Invariant Cross Attention* module to update residue embeddings while preserving local SE3 invariance.
2. Implementing an effective conditioning approach for flow-matching models: they used classifier-free guidance to interpolate flow vectors across translation, rotation, and amino acid spaces, and employ self-conditioning to refine predicted structures

Extensive experiments show that the proposed model can faithfully follow spatial conditioning, resulting in greater diversity, novelty, and improved secondary structure composition. This improves Multiflow by generating proteins with secondary structures more similar to natural proteins.
Overall, this work presents a straightforward approach to control protein generation, enhancing Multiflow's diversity, novelty, and secondary structure accuracy.

**Strengths:**

**[Clarity & Quality]**
- The manuscript is well-written, with a thorough introduction and background information on protein structure generation, spatial conditioning, and flow matching for data generation. Overall, it provides a smooth reading experience.
- The paper is of good quality, presenting clear mathematical foundations grounded in current techniques for protein modeling, diffusion-based generation, and guided sampling.
- The problem is well-defined, and the authors designed several experiments to evaluate model performance in 1) following conditioning, 2) improving general performance, and 3) demonstrating practical use in flexible conditioning, with both quantitative and qualitative comparisons.

**[Significance]**
- The spatial conditioning approach using ellipsoids is intuitive for practical applications and has potential implications for the utility of protein generation models.
- The proposed methods appear generalizable to various spatial conditioning scenarios. (*However, this paper focuses solely on secondary-structure conditioning.*)

**[Originality]**
- The authors introduce a novel conditioning modality using spatial ellipsoids for protein generation, along with a new layer, "invariant cross-attention," to integrate this information.

**Weaknesses:**

**[Clarity]**
- Some design choices and model details lack clear explanations or in-depth examination (see Q1, 2, and 4).

**[Soundness]**
- Performance on Natural Proteins: While the authors demonstrate high designability at fixed helicity levels on synthetic data, it isn't as clear if these benefits hold for natural proteins (see Q3).

**[Significance]**
- Scope: The current methods are examined only on Multiflow and for secondary structure guidance. Their practical impact on other protein generation models and types of spatial conditioning (e.g., domains, hydrophobic cores) is not extensively explored.

**Questions:**

1. **Ellipsoid Representation**

    a. Choosing the Number of Ellipsoids (k):
    - *Training:* Is k determined by the structure of the training protein? If so, what is the distribution of k in natural proteins?
    - *Evaluation based on the statistical model:* The authors appear to have used a fixed k=5 in the experiments. How was this number chosen? Have the authors tested other k values?

    b. Number of Residues per Ellipsoid:
    - The current representation specifies the number of residues in each ellipsoid, but the authors show that this number directly depends on ellipsoid volume. Could specifying the number of residues be redundant, and might removing this constraint provide the model more flexibility in generation? Have the authors examined the impact of residue count on amino acid (AA) prediction?

2. **Invariance Cross-Attention (ICA) Layer Design**

    a. The authors separately model the SE3 features (E_k) and scalar features (e_k) of ellipsoids using the proposed ICA and transformer to achieve SE3 invariance in the local frame. Has the team considered alternative approaches, such as modeling ellipsoids as “pseudo frames” with SE3 and scalar features and simply using IPA to update ellipsoid and residue features together?

    b. In Algorithm 1:
    - Could the authors clarify the *PosEmbed* used in line 223? Does it include distance, angles, or local coordinates?
    - Could they also explain why the query uses un-updated $s$, while the key and value use $a$, which incorporates current ellipsoid information?

3. **Results in Table 2 (natural proteins):** The model, even with the strongest guidance, tends to overestimate helices in proteins. Additionally, the authors did not present designability results, which, based on Figure 16, may be compromised with strong guidance. Could the authors elaborate on the model's performance in addressing the “overrepresented helix problem,” the trade-offs with other metrics, and its overall comparison to models like *RFDiffusion*?

4. In self-conditioning (line 290), they propose supplying interpolated conditions to both conditional and unconditional models, suggesting this improves “designability and ellipsoid adherence for all $\lambda$ values.” However, no ablation studies were provided to verify this claim.

---

> ### Author Response · Authors · 2024-11-17
> **Response by Authors Part 1/2**
>
> Thank you for the review! To address your questions and concerns (our updated manuscript can be downloaded at the top of this page):
>
> ---
>
> **Limitation to secondary structure conditioning**
>
> We acknowledge that conditioning on ellipsoids with functional annotations beyond secondary structure would be highly interesting. We considered the feasibility of this direction during the project using data from function annotation databases such as Interpro or using predicted annotations from models such as DeepFRI. However, the data quality is insufficient for training a model conditioned on ellipsoids with function annotations. For instance, in Interpro, many location-specific annotations are assigned to the whole protein instead of the functionally relevant site or the DeepFRI location predictions conflict with each other for different functions. Thus, we provide a general framework that allows for arbitrary annotation types but only implement it for secondary structure annotations - once sufficient quality data of broader annotations is available, the more general model can be trained.
>
> **1.a.1: "Is k determined by the structure of the training protein? If so, what is the distribution of k in natural proteins?"**
>
> We now clarify in A.1 that during training, the data determines the number of ellipsoids and added Figure 11 with a histogram of $K$.
>
> **1.a.2: $K=5$ for the synthetic ellipsoid experiments**
>
> We added a clarification in A.4 that we chose a fixed $K$ for consistency across protein lengths and the specific value of $K=5$ since it is the most frequent number of ellipsoids in PDB proteins (see Figure 11).
> We added Figure 17 to the paper where we show proteins generated with different $K$.
>
> **1.b: "The current representation specifies the number of residues in each ellipsoid, but the authors show that this number directly depends on ellipsoid volume. Could specifying the number of residues be redundant, and might removing this constraint provide the model more flexibility in generation?"**
>
> We added to the discussion in Appendix B that this is a reasonable alternative and that we chose to specify the number of residues per ellipsoid to give the user the option of controlling it (or having the number be determined via linear fit if preferred).
>
> **2.a: The authors separately model the SE3 features (E_k) and scalar features (e_k) of ellipsoids using the proposed ICA and transformer to achieve SE3 invariance in the local frame. Has the team considered alternative approaches, such as modeling ellipsoids as “pseudo frames” with SE3 and scalar features and simply using IPA to update ellipsoid and residue features together?**
>
> We added this discussion to Appendix B:
>
> We indeed considered this. However, a canonical assignment of a frame to an ellipsoid is not possible. To see this, we consider the worst-case scenario of a round ellipsoid - clearly, no canonical assignment of a 3D orientation is possible. In the best-case scenario of an ellipsoid with three distinctly sized principal components, we could choose, e.g., the two largest to construct a frame from. However, their sign is arbitrary, leading to 4 options among which no canonical choice exists. Thus, we opted for our ICA without ellipsoid token updates, which is empirically sufficient for strong ellipsoid adherence and, together with our transformer layers, aligns with the mechanisms commonly employed in computer vision [1,2,3].
>
> **2.b.1: clarifing PosEmbed in Alg. 1**
>
> We added to A.1 in the revised manuscript that PosEmbedd is a sinusoidal positional encoding of the relative positions (the vectors between ellipsoid means and residue positions). Each number of the 3D offset vector is encoded into 64 dimensions, and all 3 are concatenated.
>
>
> **2.b.2: On why the query uses un-updated s, while the key and value use a, which incorporates relative positional information**
>
> We added to Appendix B that this approach to inject relative positional encoding was our default choice since it is how relative positional encodings are used in language model transformers [4] or in geometric transformers such as "SE3-Transformer".

---

> ### Author Response · Authors · 2024-11-17
> **Response by Authors Part 2/2**
>
> **3: Results in table 2, the impact of guidance on the tradeoff between designability vs. helicity+diversity+novelty, and comparisons to RFDiffusion, Chroma, and Multiflow.**
>
> We removed our mention of "natural proteins," which may have been confusing and could leave the impression that there is a notion of comparing models on synthetic vs. natural proteins, which is not the case. All models (ProtComposer, RFDiffusion, Chroma, Multiflow) can sample a distribution of protein structures p(X) without any further inputs. In ProtComposer, this distribution is decomposed as p(X)=P(X|E)p(E) into a distribution of ellipsoids p(E) and a distribution of structures conditioned on ellipsoids P(X|E).
>
> In Figure 4 we generate proteins from scratch with all methods. Table 2 shows numbers for ProtComposer when generating from a special type of ellipsoids - ellipsoids extracted from PDB proteins. This would not be available when attempting to generate a protein from scratch.
>
> Regarding the guidance and the tradeoff: We introduce the guidance mechanism as a control knob for trading off between designability vs. helicity, diversity, and novelty. In other models, such as RFDiffusion, this tradeoff can be controlled via the sampling temperature. We compare the strength of the tradeoffs that can be achieved for all models via their tradeoff curves in Figure 4. It shows ProtComposer's improved tradeoff between designability and the other metrics, including helicity if lower helicity is the goal (note the minus sign in "1 - helicity" on the y-axis).
>
>
>
> **4: Self-conditioning via interpolation vs. other variants**
>
> We added Section C.2 with Figure 9 to the revised manuscript which shows the designability vs. ellipsoid adherence frontier under our 5 different possibilities of performing self-conditioning, showing that the interpolation variant performs best under all settings of guidance strength.
>
> ---
> We hope the discussions and additions address your concerns! When appropriate, we also reference our additions and clarifications in the main text. Please let us know if there are any further opportunities to improve the score.
>
> [1] Adding Conditional Control to Text-to-Image Diffusion Models\
> [2] GLIGEN: Open-Set Grounded Text-to-Image Generation\
> [3] Compositional Text-to-Image Generation with Dense Blob Representations\
> [4] Self-Attention with Relative Position Representations

---

> > ### Comment · Reviewer_679L · 2024-11-22
> >
> > I would like to thank the authors for their excellent work and thorough response. My concerns have been resolved and I have raised my score accordingly.

---

### Official Review · Reviewer_7wor · 2024-11-04

**Soundness:** 4
**Presentation:** 4
**Contribution:** 3
**Rating:** 8
**Confidence:** 3

**Summary:**

This paper proposes a framework to generate protein structures by conditioning on layouts specified through 3D ellipsoids. The conditions include location, size, orientation and secondary structure. These conditions are injected to flow-base protein generative models via proposed cross-attention and update modules. It shows greatly improved controllability and designability over baselines.

**Strengths:**

This paper is well motivated, formulated, written, and evaluated.

1. The injection of ellipsoid information is achieved through cross attention. This allows the ellipsoids to be unordered set. i.e. user doesn’t have to specify the order or ellipsoids; model also decides the order of ellipsoids.
2. The formulation of ellipsoid token is effective and is easy to be extended to different conditions other than secondary structure. (such as hydrophobicity)
3. Thorough analysis of ellipsoid consistency, including both geometric and probabilistic metrics.
4. Fig4: The comparison on designability/diversity with baseline methods are done as comparing Pareto frontiers with varying sampling temperatures/guidances. This clearly shows the tradeoff between the methods and the performance improvement from baselines.  I found this analysis insightful and believe other papers arguing increased performance could benefit from a similar evaluation scheme.
5. The practical use-case of the method is shown in Section4.3 flexible conditioning.
6. The controllability is greatly improved from Chroma (Table1). Accuracy and coverage is very impressive.

**Weaknesses:**

How is sequence design performance? What I understand is that the designability is solely based on the generated structure (i.e. generated sequence is discarded). Can you also present co-design designability value as in MultiFlow paper?

Other than that, I did not find any major weaknesses from the paper. However, the ablation study can be improved. Most of the model ablations are based on guidance strength, but I am also curious about ablation study on (i) ellipsoid segementation cutoff (5A currently), (ii) allow residue token to update ellipsoid token or not. For (ii), explaining the reasoning behind the design choice may suffice.

**Questions:**

Is it possible to set the order of ellipsoids? Or how complicated would it be to extend this framework to allow user to set the order of ellipsoids?

---

> ### Author Response · Authors · 2024-11-17
> **Response by Authors**
>
> Thank you for the review! To address your questions and concerns (our updated manuscript can be downloaded at the top of this page):
>
> ---
>
> **Sequence-structure co-generation results**
>
> We added Table 3 where we select statistical model parameters with high designability ($\nu = 50, \sigma=5$) and draw 400 structures together with sequences from ProtComposer. As in the Multiflow paper, the joint generation does not improve designability. Interestingly, the median self-consistency RMDSs (scRMSD) of the jointly generated sequences are worse, while their mean scRMSDs are better.
>
> 1-seq Designability refers to generating 1 sequence per structure, while 8-seq Designability uses the best out of 8 sequences per structure. Self-consistency RMSD is abbreviated as scRMSD.
>
> | Approach         | 1-seq Designability ↑ | Median scRMSD ↓ | Mean scRMSD ↓ | 8-seq Designability ↑ |
> |------------------|-----------------------|-----------------|---------------|-----------------------|
> | Joint Generation | 0.75                  | 1.76            | 2.15          | --                    |
> | ProteinMPNN      | 0.81                  | 1.65            | 2.41          | 0.98                  |
>
>
> **Ellipsoid segmentation cutoff**
>
> We added Figure 14 to the paper, which visualizes the effect of different radius cutoffs on ellipsoids. We observe that a 5A cutoff offers reasonable 3D ellipsoid segmentations that are neither too coarse nor too fine-grained.
>
> **Communication between ellipsoid and residue tokens**
>
> We added this discussion to Appendix B:
>
> In our transformer layers (algorithm 2), the residue tokens update all ellipsoid tokens, and ellipsoid tokens update all residue tokens. In Invariant-Cross-Attention (algorithm 1), we inject ellipsoid position and geometry information into the residues tokens without updating ellipsoid tokens.
>
> The reason: ICA updates a residue token based on transforming the ellipsoid means and covarianc matrices into the local coordinate frame of the residue (where residue frames are defined as in AlphaFold2). The same mechanism is not applicable to update ellipsoid tokens based on residue positions since a canonical assignment of a frame to an ellipsoid is not possible. To see this, we consider the worst-case scenario of a round ellipsoid - clearly no canonical assignment of a 3D orientation is possible. In the best-case scenario of an ellipsoid with three distinctly sized principal components, we could choose, e.g., the two largest to construct a frame from. However, their sign is arbitrary, leading to 4 options among which no canonical choice exists. Thus, we opted for our ICA without ellipsoid token updates, which is empirically sufficient for strong ellipsoid adherence and, together with our transformer layers, aligns with the mechanisms commonly employed in computer vision [1,2,3].
>
>
>
> **"Is it possible to set the order of ellipsoids?"**
>
> A simple approach to allow for setting the order would be retraining a model with an additional ellipsoid feature that encodes a list of the orders in which the ellipsoid is "hit".
>
> **Further ablation results**
>
> Since you advised adding ablations as a last avenue to improve the paper further, we also added an investigation of different self-conditioning approaches. We added Section C.2 with Figure 9 to the revised manuscript which shows the designability vs. ellipsoid adherence frontier under our 5 different possibilities of performing self-conditioning.
>
> ---
> We hope the discussions and results address your concerns! Please let us know if there are any further opportunities to improve the score.
>
> [1] Adding Conditional Control to Text-to-Image Diffusion Models\
> [2] GLIGEN: Open-Set Grounded Text-to-Image Generation\
> [3] Compositional Text-to-Image Generation with Dense Blob Representations

---

> > ### Author Response · Authors · 2024-11-25
> > **Response by Authors 2**
> >
> > With the discussion period ending tomorrow, we thank you for the work together toward a better paper!
> >
> > Since we are excited that you, as well as all other reviewers, consider our work as a good submission, please let us know if there is anything we can do to further improve the paper and make you consider modifying and raising your paper rating from “8: accept, good paper” to “10: strong accept, should be highlighted at the conference”, or if you think such a score increase is already warranted.

---

> > > ### Comment · Reviewer_7wor · 2024-11-26
> > >
> > > Thank you for thorough response to the questions. Figure 14 and your response on residue token explain the design choice well. Regarding the score, I second the response from Reviewer eiq5. I appreciate the theoretical ground of this work, the effort to demonstrate use cases, and how this conditioning can increase diversity of designs, but I will respectfully keep my score.

---

### Author Response · Authors · 2024-11-17
**Overall Response by Authors**

# Overall Response by Authors

We thank all reviewers for their constructive feedback and their time taken to review!

Next to the individual responses, we updated the manuscript that can be downloaded above with **new figures and results** and writing improvements. Changes are highlighted in blue (the final version will not contain the coloring). The main additions include:

- Section C.2 with an ablation study and Figure 9 which shows the designability vs. ellipsoid adherence frontier under our 5 different possibilities of performing self-conditioning as suggested by reviewers **D7Wa** and **679L**. Our chosen "interpolate" option is best under all guidance strengths.
- Table 3 with results for sequence-structure co-design as suggested by reviewer **7wor**. As in the Multiflow paper, co-generation does not improve designability.
- Figure 14 to  visualize the effect of different segmentation radius cutoffs on the resulting ellipsoids, showing that 5A provides a good level of granularity.
- Figure 17 showing proteins generated for different $K$ (numbers of ellipsoids) from our statistical model.
- Table 4 with compositionality results for Multiflow, Chroma, and RFDiffusion as suggested by reviewer **eiq5**.
- All suggested clarity improvements such as PosEmbed explanation (**679L**), pseudo-frames for ellipsoids (**679L**, **7wor**), user guidelines (**eiq5**), number of ellipsoids/residues per ellipsoid (**679L**), ...

---

### Meta-Review · Area_Chair_UYkf · 2024-12-20

**Metareview:**

The paper introduces ProtComposer, a framework for generating protein structures conditioned on 3D ellipsoids that encode spatial layout information. The reviewers commend the paper for being well-written and addressing a relevant, well-defined problem with practical applications. The reviewers also found the experimental evaluation to be strong and the results to be impressive.  Concerns were raised regarding the performance of natural proteins, and the introduction of a new compositionality metric also raised concerns about its validity. However, most of the reviewers' concerns about the papers were addressed with the rebuttal. The reviewers unanimously agree that this is a very strong paper, and therefore, I recommend accepting the paper.

**Additional Comments On Reviewer Discussion:**

The discussion phase mainly consisted of the rebuttal from the authors and the reviewers acknowledging the comments and changes from the authors:

* Reviewer 7wor requested co-designability results, and the authors added Table 3.
* Reviewers sought clarifications on the number of ellipsoids, and the authors added Figure 14 to illustrate segmentation radius effects and a histogram of ellipsoid counts in Figure 11.
*  Reviewers D7Wa and 679L requested ablation studies on self-conditioning approaches, and the authors added Section C.2 and Figure 9.
*Reviewer eiq5 expressed concerns about the introduced metric's validity, and the authors included comparisons to other models in Table 4 and added an explanation to Appendix B.

Overall, reviewers appreciated the thoroughness of the responses.

---

### Decision · Program_Chairs · 2025-01-22

Accept (Oral)